# Reference Intervals for Blood Biomarkers in Farmed Atlantic Salmon, Coho Salmon and Rainbow Trout in Chile: Promoting a Preventive Approach in Aquamedicine

**DOI:** 10.3390/biology11071066

**Published:** 2022-07-18

**Authors:** Marco Rozas-Serri, Rodolfo Correa, Romina Walker-Vergara, Darling Coñuecar, Soraya Barrientos, Camila Leiva, Ricardo Ildefonso, Carolina Senn, Andrea Peña

**Affiliations:** Pathovet Labs, Puerto Montt 5480000, Chile; racorrea.aipa@gmail.com (R.C.); romina.walkerv@gmail.com (R.W.-V.); darling.conuecar@pathovet.cl (D.C.); soraya.barrientos@pathovet.cl (S.B.); camila.leiva@pathovet.cl (C.L.); ricardo.ildefonso@pathovet.cl (R.I.); carolina.senn@pathovet.cl (C.S.); andrea.pena@pathovet.cl (A.P.)

**Keywords:** Atlantic salmon, coho salmon, rainbow trout, blood biomarkers, reference intervals, fish health

## Abstract

**Simple Summary:**

We report the integrated reference intervals (RIs) of 44 blood biomarkers for presmolts, smolts, postsmolts and adults of intensively farmed Atlantic salmon, coho salmon and rainbow trout species in Chile. Overall, RIs were obtained from 3.059 healthy salmon and trout from 78 different culture centers. Our results indicate that the variability of most blood biomarkers depends on the salmonid species, age range and/or interaction between them, but they are often biologically related to each other. Finally, we provide a standardized pre-analytical protocol to improve preventive vision in aquamedicine. RIs for blood biomarkers specific to salmonid species and age ranges are essential to help improve clinical, zootechnical and nutritional management for the health and welfare of farmed fish.

**Abstract:**

The mission of veterinary clinical pathology is to support the diagnostic process by using tests to measure different blood biomarkers to support decision making about farmed fish health and welfare. The objective of this study is to provide reference intervals (RIs) for 44 key hematological, blood biochemistry, blood gasometry and hormones biomarkers for the three most economically important farmed salmonid species in Chile (Atlantic salmon, coho salmon and rainbow trout) during the freshwater (presmolt and smolt age range) and seawater stages (post-smolt and adult age range). Our results confirmed that the concentration or activity of most blood biomarkers depend on the salmonid species, age range and/or the interaction between them, and they are often biologically related to each other. Erythogram and leukogram profiles revealed a similar distribution in rainbow trout and coho salmon, but those in Atlantic salmon were significantly different. While the activity of the most clinically important plasma enzymes demonstrated a similar profile in Atlantic salmon and rainbow trout, coho salmon demonstrated a significantly different distribution. Plasma electrolyte and mineral profiles showed significant differences between salmonid species, especially for rainbow trout, while Atlantic salmon and coho salmon demonstrated a high degree of similarity. Furthermore, electrolytes, minerals and blood gasometry biomarkers were significantly different between age ranges, suggesting a considerably different distribution between freshwater and seawater-farmed fish. The RIs of clinically healthy fish described in this study take into account the high biological variation of farmed fish in Chile, as the 3.059 individuals came from 78 different fish farms, both freshwater and seawater, and blood samples were collected using the same pre-analytical protocol. Likewise, our study provides the Chilean salmon farming industry with standardized protocols that can be used routinely and provides valuable information to improve the preventive vision of aquamedicine through the application of blood biomarkers to support and optimize health, welfare and husbandry management in the salmon farming industry.

## 1. Introduction

Chile is the world’s second largest producer of salmonids due to the 978.328 metric tons harvested in 2021, which consisted of 74.9% Atlantic salmon (*Salmo salar*), 19.3% coho salmon (*Oncorhynchus kisutch*) and 5.8% rainbow trout (*Oncorhynchus mykiss*) [1]. Consequently, it is essential to generate strategies to maximize the health and welfare of these farmed salmonid species in order to optimize their productivity and sustainability. Clinical pathology is a specialty of veterinary medicine that supports the diagnosis of diseases through laboratory tests using fish blood and providing necessary tools for decision making in the field of health, welfare, nutrition and, indirectly, the husbandry and environment where animals are farmed [2,3,4,5,6]. For practical reasons, in this paper we will refer to the set of hematological parameters, biochemistry and blood gases and hormones as “blood biomarkers”.

Blood is distributed to all tissues and organs of the body by the vascular system, so that the presence or absence of tissue-specific components in the blood makes it possible to determine any alterations in the tissues. Total blood volume in various species of salmonid fish has been estimated at 4.5 ± 1.5 to 7.2 ± 0.1 mL/100 g body weight using the re-injected Evan’s blue technique (4.5 to 7.2%), while total plasma volume in the same fish was measured at 3.0 ± 0.5 to 4.8 ± 1.3 mL/100 g [7]. Alterations in the number of leukocytes and their differential count (lymphocytes, neutrophils, eosinophils and monocytes) represent important clinical indicators as they point to possible infectious causes, acute and/or chronic stress, among others [6,8]. Fish neutrophils show myeloperoxidase activity in cytoplasmic granules, which makes them more similar to mammalian neutrophils than to heterophils of birds and reptiles [9]. Furthermore, basophils have only been identified in 4 out of 121 species of cartilaginous and teleost fish [10], and in 14 out of 20 species of freshwater fish [11].

Erythrocytes are the predominant blood cells in the vast majority of fish species and, unlike in mammals, fish erythrocytes and thrombocytes are nucleated cells [9]. This is the main biological explanation why clinical and research laboratories continue to use manual hematological methods. There are some reports on the use of automated methods in fish [12,13], but they have not been readily adopted. In addition, it is important to emphasize that the use of automated hematological methods must be preceded by technical validation using traditional manual analysis in each laboratory [6]. With the development of the machine learning, there are some attempts to computerize the hematological study and some guidelines for designing and evaluating them are provided [14]. Using machine learning, Gültepe and Gültepe [15] described hematological parameters of sea bream (*Sparus aurata*) and, Mani et al. [16] evaluated the use of probiotics on hematological parameters in common carp (*Cyprinus carpio*).

Blood biochemistry is based on the detection and quantification of elements such as enzymes, substrates, minerals, among others, in plasma or serum. The methods used in mammals have been adapted for fish analysis, but the results interpretation may be different as they are directly or indirectly influenced by different intrinsic factors such as fish species, productive stage, sex, nutritional, metabolic and reproductive status [3,16,17,18,19,20,21,22,23,24,25,26,27,28,29,30,31] and extrinsic factors such as environmental conditions, water type, stocking density, capture and sampling method, health status or disease [3,4,25,30,31,32,33,34].

Similarly, some parameters associated with ion-regulation, acid/base balance and hormones can be quantified in farmed salmonids, which represent important biomarkers or proxies to assess key pathophysiological processes such as smoltification, stress and reproduction [35,36]. These applications are especially relevant for salmon farmed in recirculating aquaculture systems (RAS) that use both freshwater and seawater. As a result of RAS production, CO_2_ in solution is acidic and will have an impact on both the pH and alkalinity of the system [37]. Operational recommendations for CO_2_ levels in RAS systems are typically below 15 mg/L; however, it has been demonstrated that salmonids can tolerate CO_2_ levels of 20–25 mg/L in high alkalinity freshwater without adverse effects [38,39]. Chronically elevated CO_2_ levels have often been associated with the development of nephrocalcinosis in RAS farmed fish [40,41]; therefore, blood gases biomarkers can be correlated with CO_2_ concentrations in the water where fish are cultured and ultimately with their welfare and productive performance.

Overall, all pre-analytical and analytical factors can affect the results of blood biomarkers in farmed fish, so not only is experience and care necessary to obtain reliable results, but it is critical to estimate reference intervals (RIs) representative of the distribution of each blood biomarker under normal or healthy conditions. There are some studies describing the RIs for several biochemical and hematological parameters in farmed salmonid species [29,30,42,43,44,45,46], but there is no published information on RIs for these biomarkers analyzed comprehensively and comparatively by salmonid species on a sufficiently representative sample size of fish for different age ranges and water types, and using the same pre-analytical, analytical and post-analytical procedure. Hence, the objective of this work is to provide RIs for 44 key hematological, blood biochemistry, blood gasometry and hormones biomarkers for the three most economically important farmed salmonid species during the freshwater (presmolt and smolt age range) and seawater stages (post-smolt and adult age range) in Chile, contributing to improve the interpretation and application of clinical laboratory test results in aquamedicine.

## 2. Material and Methods

### 2.1. Pre-Analytical Stage

#### 2.1.1. Fish Selection and Catching

A total of 3059 healthy fish of the three salmonid species from 78 fish farms in Chile were collected during December 2014 and May 2015, and January and June 2017 to establish RIs for the most important blood biomarkers applied to assess fish health and welfare in salmon aquaculture (Table 1). Of the fish sampled, 50.7% (1550 fish) were Atlantic salmon, 26.0% (794 fish) were rainbow trout and 23.4% (715 fish) were coho salmon. The 14.1% (432 specimens) and 24.8% (759 specimens) of the sampled fish were presmolts (<50 g) and smolts (50 to 150 g) reared in freshwater, respectively (Table 1). Presmolt and smolt individuals were collected from 14 and 20 hatcheries, respectively, located in the Araucanía, Los Ríos, Los Lagos, Aysén and Magallanes regions. Likewise, the 31.1% (952 specimens) and 29.9% (916 specimens) of the sampled fish were postsmolts (150 to 800 g) and adult fish (>800 g) reared in marine cage farms, respectively (Table 1), located in Los Lagos, Aysén and Magallanes regions. Prior to selecting each fish group, the absence of clinical disease or asymptomatic carrier status of enzootic pathogens such as *Flavobacterium psychrophilum*, *Renibacterium salmoninarum*, *Piscirickettsia salmonis*, Infectious Salmon Anemia Virus (ISAV), Infectious Pancreatic Necrosis Virus (IPNV) and Piscine Orthoreovirus (PRV) was confirmed by RT-PCR. Furthermore, the health status of each farm was certified by the veterinarian in charge by means of health, welfare, husbandry management and environmental conditions. Farms with positive or diseased fish were immediately discarded. All freshwater-reared fish (presmolt and smolts) were captured using the same protocol, regardless of salmon producer and hatchery. Briefly, no more than five fish were collected at the same time directly from each selected tank using a small fishing net with handle and then quickly deposited into the bucket with anesthesia. In addition, all sea-reared fish (postsmolt and adults) were also captured using the same protocol between seawater farms, but fish were captured by the crowd and net method.

#### 2.1.2. Anesthetic Procedure

Prior to anesthesia, a minimum of 12 h of fasting of the specimens were checked and the water conditions in the containers were controlled. Fish were exposed to a solution of 15 to 20 mL of 20% benzocaine per 100 L of water for 2 to 5 min depending on whether deep sedation (non-lethal sampling) or euthanasia (lethal sampling) was to be induced. For non-lethal sampling (>50 g), fish were immediately returned to a container with fresh water after sampling and the recovery process was monitored. Smaller animals (<50 g) were euthanized by benzocaine overdose according to animal welfare standards and, once blood samples were obtained, the fish were discarded according to the General Sanitary Program for Mortality Management of the National Fisheries and Aquaculture Service (Sernapesca).

#### 2.1.3. Blood Sampling Procedure

Whole blood samples for gasometry were collected in a volume that varied from 1 to 3 mL from the caudal vein of each fish using 1 mL capacity heparinized syringe with Pulset™ technology and Crickett™ Needle Protection (SunMed, Grand Rapids, MI, USA). Each heparinized syringe was filled to the maximum because the incomplete filling could alter the results. Air was quickly removed from the syringes and the blood was thoroughly mixed by inversion before injecting the sample into the portable equipment. Whole blood samples for hematological and blood biochemistry tests were collected in a volume that varied from 1 to 3 mL from the caudal vein of each fish using a non-vacuum sealed blood collection tube containing lithium heparin (BD, Franklin Lakes, NJ, USA). The needle was disassembled from the syringe and the blood was carefully emptied using the inner wall of the respective tube. Each tube was filled to the mark indicated by the manufacturer because the volume of blood should be commensurate with the amount of anticoagulant. The tubes were shaken gently by inversion between 10 to 15 times until the correct homogenization with the anticoagulant was achieved.

#### 2.1.4. Procedure for Transport, Preparation and Storage of Samples at the Laboratory

The tubes were labeled and placed in expanded polystyrene boxes with gelpack or ice and datalogger to transport the samples to the laboratory under a carefully maintained cold chain. Part of the whole blood volume was used for hematological analysis and the rest was centrifuged at 10,000× *g* for 3 min to separate the plasma, which was transferred to a labeled tube and placed on wet ice at 4 °C until analysis. All samples were analyzed 24–48 h after blood collection.

### 2.2. Analytical Stage

#### 2.2.1. Hemotological and Blood Gasometry Biomarkers

The blood gas biomarkers analyzed were bicarbonate ion concentration (HCO_3_), partial pressure of carbon dioxide (pCO_2_) and hydrogen potential (pH). Heparinized whole blood samples were analyzed for blood gasometry using the IRMA TRUPOINT^®^ System single-use Point-of-Care (POC) analyzer (Lifehealth, Roseville, MN, USA). The cartridges were removed from their packaging, their protective tape was removed, and they were fully inserted into the analyzer. Once the cartridge was inserted, the equipment was automatically calibrated and the whole blood sample was injected directly from the heparinized syringe. Once the analysis of each sample was completed, the cartridge and syringe were removed and discarded in a biological material container. The results were automatically displayed on the touch screen of the equipment when the analysis was finished and the voucher with the printed results was obtained.

The blood count biomarkers analyzed were hematocrit (Htc), red blood cell count (RBC), hemoglobin (Hgb), mean corpuscular volume (MCV), mean corpuscular hemoglobin concentration (MCHC), white blood cell count (WBC), lymphocytes (LYM), neutrophils (NEU), monocytes (MON) and thrombocyte count (TCC). Hemoglobin concentration was estimated by the cyanomethemoglobin method using the HumaMeterTM Hb Plus (HUMAN, Wiesbaden, Germany), while Hct was determined by centrifuging the microcapillary-loaded blood at 10,000× *g* for 10 min at room temperature in a Frontier™ 5515 microcentrifuge (Ohaus, Parsippany, NJ, USA). A Neubauer hemocytometer was used to determine the total RBC, total WBC, and TCC in blood mixed with Natt–Herrick staining solution. Calculation of RBC was performed using the number of counted cells, number of squares in which they were counted, square volume and blood dilution (RBC (mm^3^) = cells counted × 5 × 10 × dilution factor). Similarly, the total WBC count per mm^3^ was determined using the number of counted cells, blood dilution, area in which they were counted and depth. The differential leukocyte count was calculated from the analysis of Giemsa-stained blood smears in which the number of various types of leukocytes per 100 cells was counted in several fields of a smear [11]. Finally, MCV and MCHC were calculated accordingly [47].

#### 2.2.2. Blood Biochemistry and Hormones Biomarkers

Plasma samples were analyzed for plasma substrates [Total protein (TPO), Albumins (ALB), Globulins (GLO), Total bilirubin (TBI), Direct bilirubin (DBI), Creatinine (CRE), Glucose (GLU), Lactate (LAC), Urea (URE), Uric acid (UAC), Ammonia (NH3), Total Cholesterol (TCH), Triglycerides (TRG), High-density lipoprotein cholesterol (HDL), and Low-density lipoprotein cholesterol (LDL)], enzymes [Alkaline Phosphatase (ALP), Alanine transaminase (ALT), Aspartate aminotransferase (AST), Total amylase (TAM), Lipase (LIP), Creatine Kinase total (CKT), Cardiac Creatine Kinase isoenzyme (CK-MB), and Lactate dehydrogenase (LDH)], electrolytes and minerals [Sodium (Na), Potassium (K), Chloride (Cl), Calcium (Ca), Magnesium (Mg), Iron (Fe) and Phosphorus (P)] using a cobas c311 autoanalyzer (Roche Diagnostics, Risch-Rotkreuz, Switzerland), while plasma cortisol (COR) concentration was determined using a cobas e411 automatic endocrinology analyzer (Roche Diagnostics, Risch-Rotkreuz, Switzerland). A standard kit developed by the manufacturer (Roche Diagnostics, Mannheim, Germany) was used in each assay.

### 2.3. Post-Analytical Stage

#### 2.3.1. Differences between Salmon Species and Age Ranges

Significant differences were evaluated for the 44 biomarkers between age range (n_i_ = 4) and salmonid species (n_j_ = 3). For this, linear models were constructed for each blood biomarker to evaluate the salmonid species and age ranges interaction. To evaluate the assumption of normality of the residuals, we used Q-Q plots and the Shapiro–Wilks test (*p* > 0.05), and for homoscedasticity we evaluated the trend of the standardized Pearson residuals and the predicted values and Leven Test (*p* > 0.05). In case the assumptions of normality and homoscedasticity of the residuals were not met, a normality adjustment with Box-Cox, Power Box-Cox, log with offset (log[Y + z]), square root (√Y) transformations were performed. Tukey’s multiple comparisons (α = 0.05) were performed between the salmonid species and age range interaction. In case a normal distribution could not be fitted, a Kruskal–Wallis test was performed between salmonid species and age ranges using Dunn’s test (Bonferroni) multiple comparison. The package “multcompView”, “Car” and “MASS” implemented in the R program (v2021.09.2) (Core Team, 2019, Vienna, Austria) were used.

#### 2.3.2. Reference Intervals (RIs) and Confidence Intervals (CIs)

The RIs and CIs for the salmonid species and age ranges interaction for all 44 blood biomarkers were estimated accordingly, as previously described by [48]. The distribution of each blood biomarker was determined by the Shapiro–Wilks test (*p* > 0.05) and the representation of the observations by Boxplot and histograms with the density distribution. Horn and Dixon method was used to determine and eliminate outliers from parametric and nonparametric distributions, respectively [49]. The package “referenceIntervals” was used to calculate the confidence intervals of the RIs according to the type of normal or non-normal distribution and bootstrap methods for small samples (n < 120) [49]. All statistical analyses were performed with the “referenceIntervals” package in the R program (R Development Core Team).

#### 2.3.3. Multivariate Analysis

To simplify the multivariate interactions between blood biomarkers and salmonid species and age ranges, principal component analysis (PCA) was performed grouped by blood biomarker types (erythrogram, leukogram, plasma substrates, plasma enzymes, plasma electrolytes, minerals and gases). Furthermore, the multivariate similarity of blood biomarkers between salmonid species and age ranges was evaluated by analysis of similarities (ANOSIM) using a Bray Curtis matrix and the groups were plotted with a non-metric multidimensional scaling (nMDS) ordination. The Vegan package implemented in R was used.

## 3. Results

### 3.1. Erythrogram

Some erythrogram biomarkers exhibited a normal distribution (e.g., Htc) or were transformed to a normal distribution (e.g., RBC, Hgb, MCV, MCHC) over the age range and/or salmonid species (Table 2, Appendix A). While MHCH demonstrated an association with age range independently of salmonid species (r < 0.20; *p* < 0.05) (Appendix A), and Htc and RBC had an association with species regardless of age range (r < 0.20; *p* < 0.05) (Appendix A), Hgb and MCV showed association with the interaction between salmonid species and age range. A cluster between RBC and Hgb was observed (Figure 1). The 56.3% of the total variability of the erythrogram profile was captured by two-dimensional analysis both between salmonid species and age range (Figure 1), mainly driven by Htc, Hgb and RBC. Furthermore, the erythrogram profile showed significant differences between species (R_ANOSIM_ = 0.3360; *p* = 0.0110), mainly contributed by Atlantic salmon, since rainbow trout and coho salmon demonstrated a high degree of similarity (Figure 1). On the other hand, erythrogram biomarkers demonstrated a high similarity between age ranges (R_ANOSIM_ = 0.1611; *p* = 0.0001), but although a homogeneous distribution was observed between smolts and post-smolt, a greater distance was detected between presmolt and adult (Figure 1). Taken together, these results reveal that the outcome of most erythrogram parameters depends on the salmonid species, age range and/or the interaction between them, and they are often biologically related to each other.

### 3.2. Leukogram

Several leukogram biomarkers exhibited a normal distribution or were transformed to a normal distribution (e.g., WBC, LYM, NEU) (Table 3, Appendix A). In addition, the MON count showed a normal distribution in coho salmon and Atlantic salmon, but not in rainbow trout, whereas the TC count demonstrated a normal distribution in Atlantic salmon and rainbow trout, but not in coho salmon (Table 3, Appendix A). The RI for the eosinophils count could not be calculated due to the lack of variability in the data. In the postsmolt and adult age range (seawater), higher levels of NEU were observed in coho salmon as well as lower counts of LYM and WBC were observed in presmolt and smolt (freshwater) of coho salmon and rainbow trout, respectively (Table 3). The WBC count had a positive association with the MON count in all salmonid species (Appendix A), but the LYM count demonstrated a positive association with the NEU count only in Atlantic salmon and with the MON count in both Atlantic and coho salmon (Appendix A). The LYM count presented a significant negative association with the NEU and MON count in rainbow trout (r < 0.20; *p* < 005) (Appendix A), but only demonstrated a significant positive association in post-smolt with the WBC, MON and NEU (Appendix A). Interestingly, each of the leukogram biomarkers demonstrated a significant positive association with each other only in the post-smolt stage (Appendix A). The 62.1% of the total variability of the leukogram profile was captured by two-dimensional analysis, both considering salmonid species and age range, mainly driven by WBC and LYM counts (Figure 2). A cluster between MON and WBC was found (Figure 2). Leukogram biomarkers were significantly different between salmonid species (Figure 2), as while we found a similar distribution between rainbow trout and coho salmon, an unrelated distribution in Atlantic salmon was detected (R_ANOSIM_ = 0.3703; *p* = 0.0001) (Figure 2). On the other hand, leukogram biomarkers profile demonstrated a high similarity between age ranges (R_ANOSIM_ = 0.3138; *p* = 0.0001), but although a homogeneous distribution was observed between smolts and post-smolt, a greater distance was detected between presmolt and adult (Figure 2). Taken together, our results denote that the level of most leukogram biomarkers depends on the salmonid species, age range and/or the interaction between them, and they are often biologically related to each other.

### 3.3. Plasma Substrates

Plasma substrates such as TPO, ALB, DBI, URE and LDL demonstrated a normal distribution or were transformed to a normal distribution (Table 4, Appendix A), while GLO, TBI, CRE, GLU, LAC, UAC, NH3, TCH, TRG and HDL did not demonstrate a normal distribution (Table 4, Appendix A). Higher levels of TPO, ALB, GLU, HDL were observed in the postsmolt and adult age range of Atlantic salmon, while in presmolt and smolt higher levels of HCT were found in Atlantic salmon and lower levels of CRE in rainbow trout. (Table 4, Appendix A). While LDL had a positive association with species independent of age range, DBI demonstrated no association with the interaction between species and age range. The plasma concentration of TPO demonstrated a significant positive association with the concentration of ALB and GLO in all three species and age ranges, as did ALB quantities with those of GLO and LDL (r < 0.20; *p* < 0.05) (Appendix A). In coho salmon, a significant positive association was observed between TCH with TPO, LDL, ALB and GLO, as well as HDL with TRG (Appendix A). In rainbow trout, TCH had a significant positive association with ALB, GLO, LDL, UAC and CRE; while CRE demonstrated a positive association with TRG, LDL, HDL and UAC (Appendix A). In Atlantic salmon, a positive association of TCH with TPO, GLO and URE was observed, but a negative association with HDL (Appendix A). LAC demonstrated a significant negative association with TPO, ALB, GLO, CRE and GLU in all age ranges of the three salmonid species, as well as NH3 with TPO, ALB, GLO and CRE, and HDL with TCH (r > −0.20; *p* < 0.05) (Appendix A). The 32.5% of the total variability of the substrates profile was captured by two-dimensional analysis, mainly driven by the TCH, HDL, TBI, TRG, URE and NH3 (Figure 3). Clusters between TPO, GLO and ALB; between TCH, LDL, LAC and TRG; and between NH3 and URE were distinguished (Figure 3). The substrate biomarker profile demonstrated significant differences between species (R_ANOSIM_ = 0.1580, *p* = 0.011) and age range (R_ANOSIM_ = 0.0910; *p* = 0.0001), although the distribution of observations was homogeneous among the different growth stages in the three salmonid species (Figure 3). These results indicate that the concentration of plasma substrates depends on salmonid species, age range and/or the interaction between them, but the variability of the distribution of the same biomarkers is more significantly associated with age range than with salmonid species.

### 3.4. Plasma Enzymes

As for plasma enzymes, only ALT, AST and CKT demonstrated a normal distribution or were transformed to a normal distribution (Table 5, Appendix A), while ALP, TAM, LIP, CK-MB and LDH were not normally distributed (Table 5, Appendix A). The highest TAM activity was observed in postsmolt and adult of coho salmon, whereas the highest CK-MB activity was recorded in presmolt and smolt of Atlantic salmon. The lowest ALP activity was observed in coho salmon independent of age range, while a similar CKT activity between species and age range was observed. Three salmonid species demonstrated the same positive association profile of ALT activity with AST, AST with LDH, LIP with TAM and CK-MB with CKT (r < 0.20; *p* < 0.05) (Appendix A), as well as the same negative association profile of LIP activity with CK-MB and CKT (r > −0.20; *p* < 0.05) (Appendix A). Interestingly, the positive association of AST activity with LDH, CKT and CK-MB is common in coho and Atlantic salmon, whereas in rainbow trout the positive association was observed with ALP, LIP and TAM activity (r < 0.20; *p* < 0.05) (Appendix A). Similarly, ALT activity in coho and Atlantic salmon showed a positive association with AST, LDH and LIP, but it was positively associated with the activity of LDH, ALP, CKT, AST and TAM in rainbow trout (r < 0.20; *p* < 0.05) (Appendix A). A positive association between CK-MB activity with CKT, LDH and ALP was observed only in Atlantic salmon (r < 0.20; *p* < 0.05) (Appendix A). ALT demonstrated a significant positive association with AST in smolt and postsmolt, but not in presmolt or adults (Appendix A). Similarly, LIP presented a significant positive association with TAM in smolt and adult, but not in presmolt and postsmolt (Appendix A), while CK-MB revealed a significant positive association with other important enzymes of skeletal and cardiac muscle tissue such as ALP, AST and CKT in smolt and postsmolt (Appendix A). The 33.7% of the total variability of the plasma enzymes profile was captured by two-dimensional analysis both salmonid species and age ranges, mainly driven by TAM, CK-MB and CKT (positive) and AST and ALP (negative) (Figure 4). Clusters were observed between ALP, AST and LDH, and between TAM, CK-MB, CKT, LIP and ALT (Figure 4). The enzyme profile demonstrated significant variability between species (R_ANOSIM_ = 0.2109, *p* = 0.0001), mainly attributed to Atlantic salmon, since the distribution of observations was similar between rainbow trout and coho salmon (Figure 4). Regarding age range, the enzyme profile showed low variability (R_ANOSIM_ = 0.07708, *p* = 0.0001), explained by a uniform distribution between presmolt and smolt, although distanced from postsmolt and adults (Figure 4). Taken together, our results demonstrate that the activity of most plasma enzymes depends on the salmonid species, age range and/or interaction between them, and they are often biologically related to each other according to the functionality of fish systems, organs and tissues.

### 3.5. Plasma Electrolytes and Minerals, Cortisol and Blood Gases

All plasma minerals analyzed demonstrated a normal distribution (e.g., Ca, P, Mg and Fe) (Table 6, Appendix A), whereas plasma electrolytes in coho and Atlantic salmon (e.g., Na, K and Cl) and COR did not have a normal distribution (Table 6, Appendix A). Biomarkers of blood gasometry and COR did not demonstrate a normal distribution (Table 6, Appendix A). While the median plasma cortisol concentration was similar in the smolts of the three salmonid species (34.69, 39.28 and 39.17 ng/mL in coho salmon, Atlantic salmon and rainbow trout, respectively), the median cortisol was significantly different in postsmolts among salmonid species (77.65, 55.60 and 43.26 ng/mL in coho salmon, Atlantic salmon and rainbow trout, respectively). In adult fish, median plasma cortisol was similar in coho salmon and rainbow trout (30.90 and 30.33 ng/mL, respectively), but significantly higher in Atlantic salmon (59.10 ng/mL) (Table 6). The highest plasma Na and Cl concentrations were observed in postsmolt and adult Atlantic salmon (Appendix A). A positive association between Na, Cl, Mg, Fe, K and Ca (r < 0.20; *p* < 0.05) and a negative association between Cl and K was observed (r > −0.20; *p* < 0.05) (Appendix A). Interestingly, a significant positive association between COR with Na and with Cl was noted only in Atlantic salmon (r < 0.20; *p* < 0.05) and essentially between freshwater and saltwater age ranges (Appendix A). Biomarkers of blood gasometry did not show a normal distribution (Table 6, Appendix A). Likewise, the highest pCO_2_, HCO_3_ and Mg concentrations were recorded in presmolt and smolt regardless salmonid species (Appendix A). A positive association between HCO_3_ and pCO_2_ was detected in all salmonid species (r < 0.20; *p* < 0.05) and a negative association between pH and pCO_2_ especially in coho salmon (r > −0.20; *p* < 0.05) (Appendix A). Likewise, pH demonstrated a positive significant association with HCO_3_, but a negative significant association with pCO_2_ in presmolt and smolt (Appendix A). Table 6. Reference intervals (RIs) for plasma electrolytes, minerals, cortisol, and blood gases parameters in presmolt and smolt, postsmolt and adult Atlantic salmon, coho salmon and rainbow trout reared in Chile. The respective confidence intervals (CIs) for the respective RIs are included. Letters indicate significant differences between age ranges (*p* < 0.05).

The 36.4% of the total variability of the plasma electrolytes and minerals profile was captured by two-dimensional analysis both in salmonid species and age ranges (Figure 5), mostly driven by a cluster between Na and Cl, and Mg and P. On the other hand, the 70.8% of the total variability of the blood gasometry profile was captured by two-dimensional analysis considering salmonid species and age ranges (Figure 6). The plasma electrolytes and minerals profile demonstrated significant differences between salmonid species (R_ANOSIM_ = 0.4227; *p* = 0.0001), attributed particularly by rainbow trout because the electrolytes and minerals profile in Atlantic salmon and coho salmon demonstrated a high degree of similarity. Furthermore, significant differences between age ranges were found (R_ANOSIM_ = 0.1997; *p* = 0.0001), demonstrating an appreciably different distribution between freshwater and seawater-farmed fish (Figure 5). Blood gasometry biomarkers were significantly different between salmonid species (R_ANOSIM_ = 0.2039; *p* = 0.0001) and age ranges (R_ANOSIM_ = 0.5876; *p* = 0.0001), suggesting a considerably different distribution of the gases profile between freshwater and seawater-farmed fish (Figure 6). In general, the Na and Cl concentration increases during the smolt and post-smolt stages associated with the physiological preparation of fish for transfer to seawater commanded by cortisol and gas concentration decreases as fish are cultured in seawater. The distributions of all blood biomarkers and the correlograms based on salmonid species and age ranges are proved in Appendix A.

## 4. Discussion

The mission of veterinary clinical pathology is to support the diagnostic process by using tests to measure different blood biomarkers to support decision making about farmed fish health and welfare. Consequently, here we report for the first time and with an integrated approach the RIs for 44 different blood biomarkers only from healthy individuals of the three salmonid species and age ranges farmed in Chile. Forty-one of the 44 blood biomarkers analyzed in this study changed significantly with age range (and consequently with salinity), salmonid species and/or their interaction. Taken together, our results confirm that fish growth over the production cycle, and especially the change from freshwater (smolt) to seawater environment (post-smolt and adult), is especially critical in the differences found in the different blood biomarker profiles. In particular, the smolt stage, where one of the major physiological, morphological and behavioral changes in anadromous salmonid species occur, is the turning point for several of the biomarkers.

Blood biochemistry is based on the quantification of different elements such as enzymes, substrates, minerals, electrolytes, hormones, among others. The methods used in mammals have been adapted for aquatic animals, but the interpretation of the results may be different since they are directly or indirectly influenced by intrinsic (e.g., species, age range, sex, nutritional and reproductive status, etc.) and extrinsic factors (e.g., stress, environmental conditions, population density, catching and sampling methods, etc.). This variability supports the need to estimate RIs with respect to the normality of the indicators in fish under productive conditions, according, at least, to the species and age range. An RI corresponds to a range within which the values of a biological variable are found in the majority of individuals (95%) of a clinically healthy population [48]. Population-based RI is one of the most widely used laboratory tools in the clinical decision-making process. Technically, each laboratory must generate its own RIs according to its specific pre-analytical, analytical and post-analytical procedures, but high costs are a critical limitation to its practical implementation. Although the analytical phase of clinical biochemistry is usually well controlled in laboratories, it is known that pre-analytical technical variables can influence analyte concentrations or activities.

Although there are some reports that can be compared, at least in part, with our study in different species of salmonids [19,20,21,25,29,30,31,42,46,50,51,52,53], our study is the first one that has incorporated the greatest biological variability for the estimation of RIs. The lack of a standardized protocol for blood collection and processing prior to analysis has historically limited the practical application of blood biochemistry in clinical diagnosis and disease monitoring in aquaculture, as it has generally resulted in the generation of RIs that are too wide between the maximum and minimum range, even those estimated from healthy fish. There are some studies that have investigated a range of pre-analytical treatments such a tube type, time to obtain serum or plasma, storage time and temperature, and freeze–thaw cycles, sampling methods, among others, to examine their effect on analyte concentrations and activities [6,30,46]. However, the standardization of protocols has not been an easy task, because salmonid species have also demonstrated high individual variation in blood biomarkers, and each laboratory calculates RIs based on the fish samples that come into the laboratory precisely for analysis; therefore, they are often not completely healthy. To this analysis we must add logistical and operational variables related to the fact that salmon farms are often geographically distant from established clinical laboratories and not all have the same practical conditions, starting with trained personnel, for the execution of pre-analytical protocols under strictly equal conditions.

Although previous studies have demonstrated that many preanalytical technical treatments lead to statistically significant differences in Atlantic salmon, often not large enough to be clinically significant [46], clinical practice demonstrates the opposite when RIs are calculated considering a representative biological variability. For example, our results for adult Atlantic salmon demonstrate differences from the RIs described by [46] for almost all enzymes, substrates and minerals. Thus, while our RIs were broader for LIP, PTO, ALB, GLO, Na, Cl, K, P, Mg and Ca, they were narrower for CRE, ALP, ALT, AST, CKT and LDH. That is, our RIs for substrates and minerals were consistently broader, while for enzymes they were consistently narrower. The only biomarkers that presented similar results were TAM, TCH and GLU. The variation in the results of some enzyme assays between laboratories could be related to the use of different methodologies, different manufacturers of diagnostic kits, among others, so each diagnostic laboratory should generate the RIs based on their own methods and equipment.

Blood biomarkers profiling in salmonids has been used most frequently in the characterization of infectious diseases, defining and quantifying systemic functional profiles such as liver, renal, cardiac, pancreatic and other functions [54,55,56,57,58,59,60,61,62,63,64,65,66,67,68,69]. Furthermore, the number and percentages of the different types of leukocytes present have been demonstrated to change in response to infection and stress in salmonids [29,30,70,71,72,73,74,75,76,77,78,79,80].

In fish, COR is the predominant glucocorticoid released as part of the primary stress response, and is critical in mediating metabolic, physiological and behavioral adaptive adjustments [81], which can impact health and welfare, and threaten aquaculture sustainability [82]. Recently, stress-induced cortisol production has been demonstrated to be associated with an altered gut microbiome in Atlantic salmon, specifically with a marked decrease in lactic acid bacteria *Carnobacterium* sp. and an increase in the abundance of *Clostridia* and *Gammaproteobacteria* [83]. Plasma cortisol concentration in farmed fish is a validated and recognized biomarker for monitoring fish welfare and health, but often exhibits high biological variability when an adequate and standardized sampling and sample collection protocol is not used. The plasma COR concentration in our study was significantly higher in seawater than in freshwater regardless of species, although the highest concentration was observed in Atlantic salmon. Overall, these values are slightly higher than those described in salmonids in experimental studies [84,85], included during the smoltification process [86], but are similar to those described in field conditions [30]. Under field conditions in Chile, higher plasma cortisol concentrations have been described in Atlantic salmon infested by parasitic copepod *Caligus rogercresseyi* [87].

The osmoregulatory process is essential for salmonid fish survival specially after seatransfer [88]. The concentration of Na and Cl increased with age range from freshwater to seawater in Atlantic salmon, but the concentration of Mg decreased with age range from freshwater to seawater. These results are consistent with previous reports in Atlantic salmon, which are associated with the salinity challenge experienced by the fish post-entry to the sea [35]. Blood gasometry biomarkers (pCO_2_, HCO_3_ and pH) were found significantly higher in freshwater regardless of the salmonid species. Atlantic salmon farmed on RAS and exposed to CO_2_ for 12 weeks demonstrated significantly increased blood pH, K, HCO_3_ and PCO_2_ and decreased plasma Na and Cl concentrations, suggesting that CO_2_ concentrations below the 15 mg/L threshold continue to impact Atlantic salmon [89]. 

Our results demonstrated that MON and NEU counts and Hct were found to be significantly higher in seawater regardless of the salmonid species; the LYM counts, MCHC, MVC and Hgb were significantly higher in freshwater, also regardless of the species. In rainbow trout, RIs for red blood cell indices, substrates, serum enzymes and electrolytes for freshwater farmed fish have been established [29]. PCA revealed that certain serum components were effectively differentiated between fish life stages (92.7% of the variance) than hematological principal components (80% of the variance). On contrary, our results demonstrated that while leukogram and erythrogram biomarkers captured between 62.4 and 85.1% of the variance, respectively, the blood biochemistry biomarkers captured between 46.3% to 77.1% of the variance. PTO, ALB, GLO and BTI in Atlantic salmon, DBI in rainbow trout, and NH_3_ in coho salmon were higher in seawater, but TCH and TGR were higher in Atlantic salmon and rainbow trout during the freshwater stage. This is interesting, given that TCH is crucial for the biosynthesis of steroid hormones such as cortisol [90], and has also been demonstrated to increase both the enthalpy and entropy of activation for Na^+^ K^+^ enzyme activity during the Atlantic salmon smoltification process [91]. The interaction between TCH and COR is especially important in the context of Atlantic salmon seawater adaptation [92].

Regarding the plasma enzymes, ALP was found to be significantly lower in coho salmon regardless of the age range. ALT, AST and CK-MB were found significantly higher in freshwater regardless of the salmonid species, but TAM and LIP were observed at higher levels in seawater. There are several studies reporting that body size influences blood biomarkers in different salmonid species [21,26,29,30], and there are even findings showing that in rainbow trout farmed in Italy the values of RBC, Hct, TCH and TPO were significantly lower than in Turkish trout, although no significant differences were found for WBC and Hgb [93]. In fact, it has recently been reported that between 1 and 13% of 80 Atlantic salmon farms tested during a seawater grow-out between 2017 and 2019 in Scotland demonstrated suspected anemia or clinical anemia, suggesting an eventual association with blood loss from the gills [94].

## 5. Conclusions

RIs for salmonid species- and age-range-specific blood biomarkers are essential to help improve clinical, husbandry and nutritional management for farmed fish health and welfare. Our results indicate that variability in most blood biomarkers depends on salmonid species, age range and/or interaction between them, but they are often biologically related to each other. The establishment of the RIs in this study is valuable, not only because little work has been conducted previously in farmed salmonid species, but also because no previous work has presented normal RIs for 44 blood biomarkers considering more than 3000 individuals of different age ranges and from three different species with high commercial value that came from 78 different farms. Our study provides the Chilean salmon industry with standardized protocols for routinely sampling their fish populations, since a standardized system for sampling methods and sample processing is essential for implementing comprehensive fish health monitoring. To our knowledge, this is the reference interval study for blood biomarkers that has been performed with the highest number of fish and farms of different salmonid species and age ranges using the same pre-analytical protocol and described comprehensively in the same report. The practical application of these blood biomarkers under a preventive medicine vision contributes concretely to the technical support of strategic decision making that optimizes the health and husbandry management of salmon farmers, veterinarians and health managers.

## Figures and Tables

**Figure 1 biology-11-01066-f001:**
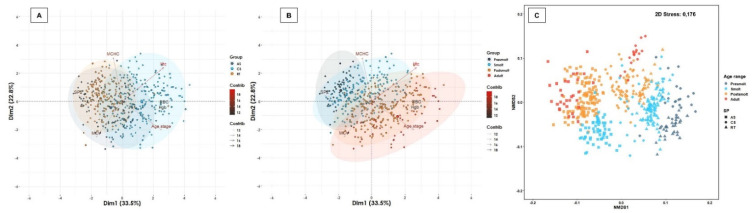
Multivariate analysis of erythrogram biomarkers. Salmonid species and range contribute significantly to the total variation of the erythrogram profile. (**A**) Spatial sorting of the erythrogram biomarkers according to salmonid species and (**B**) age ranges. The two-dimensional analysis captures 56.3% of the total variance of the erythrogram biomarker profile. Htc, Hgb and RBC contribute significantly to the total variance of the erythrogram in dimension 1, while the MCV, MCHC and Htc are biomarkers that contribute the most to dimension 2. A cluster between Hgb and RBC, MCHC and MCV, and Htc was observed. (**C**) The multivariate analysis of interdependence of the erythrogram profile demonstrates significant differences between salmonid species (R_ANOSIM_ = 0.3360; *p* = 0.0110), suggesting a uniform distribution between rainbow trout and Atlantic salmon, but another distribution for coho salmon. Similarly, significant differences were observed between age ranges (R_ANOSIM_ = 0.1611; *p* = 0.0001), suggesting differences in erythrogram biomarkers between freshwater and seawater productive stages. R_ANOSIM_ close to 0 suggests a uniform distribution of high and low ranges within and between groups, and an R_ANOSIM_ close to 1 suggests dissimilarity between groups.

**Figure 2 biology-11-01066-f002:**
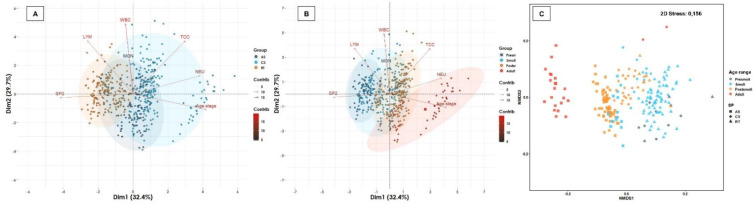
Multivariate analysis of leukogram biomarkers. Salmonid species and range contribute significantly to the total variation of the leukogram profile. (**A**) Spatial sorting of the leukogram biomarkers according to salmonid species and (**B**) age ranges. The two-dimensional analysis captures 62.1% of the total variance of the leukogram biomarker profile. NEU count contributes significantly to the total variance of the leukogram in dimension 1, while the WBC, TCC and LYM are biomarkers that contribute the most to dimension 2. A cluster between Hgb and RBC, MCHC and MCV, and Htc was observed. (**C**) The multivariate analysis of interdependence of the leukogram profile demonstrates significant differences between salmonid species (R_ANOSIM_ = 0.3703; *p* = 0.0001), suggesting a uniform distribution between rainbow trout and Atlantic salmon, but another distribution for coho salmon. Similarly, significant differences were observed between age ranges (R_ANOSIM_ = 0.3138; *p* = 0.0001), suggesting differences in leukogram biomarkers between freshwater and seawater-reared fish.

**Figure 3 biology-11-01066-f003:**
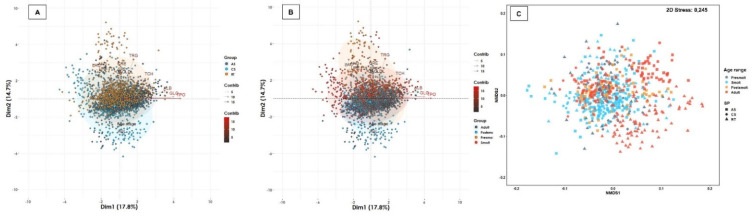
Multivariate analysis of plasma substrates biomarkers. Salmonid species and range contribute significantly to the total variation of the plasma substrates profile. (**A**) Spatial sorting of the plasma substrates biomarkers according to salmonid species and (**B**) age ranges. The two-dimensional analysis captures 54.2% of the total variance of the plasma substrate biomarkers profile. TPO, URE, NH_3_, GLO, ALB, TCH and HDL contribute significantly to the total variance of the plasma substrates in dimension 1, while the TRG is the biomarker that contributes the most to dimension 2. (**C**) The multivariate analysis of interdependence of the plasma substrates profile demonstrates significant differences between salmonid species (R_ANOSIM_ = 0.1580; *p* = 0.0110), suggesting a uniform distribution between all three salmonid species. At the same time, significant differences were observed between age ranges (R_ANOSIM_ = 0.0910; *p* = 0.0001), suggesting differences in plasma substrate biomarkers between freshwater and seawater-reared fish.

**Figure 4 biology-11-01066-f004:**
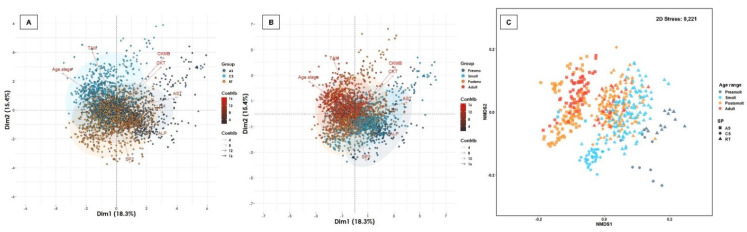
Multivariate analysis of plasma enzymes biomarkers. Salmonid species and range contribute significantly to the total variation of the plasma enzymes profile. (**A**) Spatial sorting of the plasma enzymes according to salmonid species and (**B**) age ranges. The two-dimensional analysis captures 33.7% of the total variance of the plasma substrate biomarkers profile. CKT, CK-MB and AST contribute significantly to the total variance of the plasma enzymes in dimension 1, while the ALP and TAM are the biomarkers that contributes the most to dimension 2. A cluster was observed between ALP, AST and LDH, and TAM, CK-MB, CKT, LIP and ALT. (**C**) The multivariate analysis of interdependence of the plasma enzymes profile shows significant differences between salmonid species (R_ANOSIM_ = 0.2109; *p* = 0.0001), suggesting a uniform distribution between Atlantic salmon and rainbow trout, but a different distribution in coho salmon. Concurrently, no significant differences were observed between age ranges (R_ANOSIM_ = 0.0771; *p* = 0.083), suggesting a similar distribution of plasma enzymes between freshwater- and seawater-reared fish.

**Figure 5 biology-11-01066-f005:**
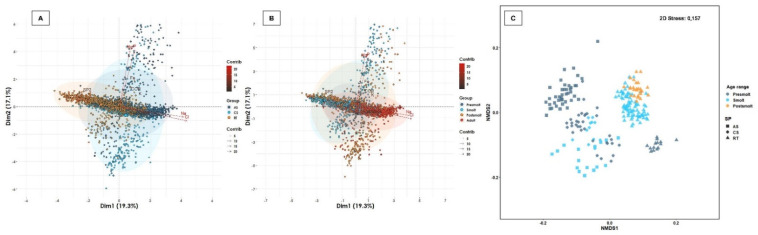
Multivariate analysis of plasma electrolytes and minerals biomarkers. Salmonid species and range contribute significantly to the total variation of the plasma electrolytes and minerals profile. (**A**) Spatial sorting of the plasma electrolytes and minerals according to salmonid species and (**B**) age ranges. The two-dimensional analysis captures 36.4% of the total variance of the plasma substrate biomarkers profile. Na and Cl contribute significantly to the total variance of the plasma enzymes in dimension 1, while the Na, Cl, P and Mg are the biomarkers that contribute the most to dimension 2. A cluster between Na and Cl, and Mg and P was detected. (**C**) The multivariate analysis of interdependence of the plasma electrolytes and minerals profile demonstrates significant differences between salmonid species (R_ANOSIM_ = 0.4227; *p* = 0.0001), suggesting a different distribution for each salmonid species. In addition, significant differences between age ranges were found (R_ANOSIM_ = 0.1997; *p* = 0.0001), suggesting a very different distribution of plasma electrolyte and mineral profile between freshwater and seawater-farmed fish.

**Figure 6 biology-11-01066-f006:**
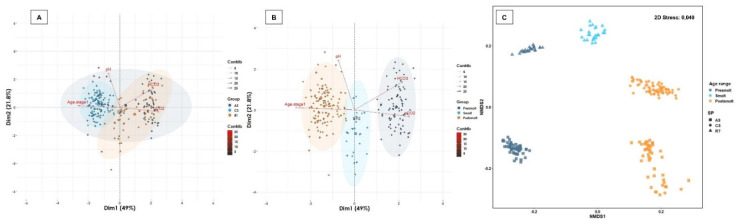
Multivariate analysis of blood gases. Salmonid species and range contribute significantly to the total variation of the blood gases profile. (**A**) Spatial sorting of the plasma electrolytes and minerals according to salmonid species and (**B**) age ranges. The two-dimensional analysis captures 70.8% of the total variance of the blood gases profile. HCO_3_ and pCO_2_ contribute significantly to the total variance of the blood gases in dimension 1, while the pH is the biomarker that contributes the most to dimension 2. A cluster between HCO_3_ and pCO_2_, and pH was detected. (**C**) The multivariate analysis of interdependence of the blood gases profile demonstrates significant differences between salmonid species (R_ANOSIM_ = 0.2039; *p* = 0.0001), suggesting a uniform distribution between Atlantic salmon and rainbow trout, but a different distribution in coho salmon. In addition, significant differences between age ranges were found (R_ANOSIM_ = 0.5876; *p* = 0.0001), proposing a different distribution of blood gasometry profile between freshwater and seawater-farmed fish.

**Table 1 biology-11-01066-t001:** Number of samples and farms to determinate RIs for blood biomarkers by salmonid species (Atlantic salmon, coho salmon, and rainbow trout) and age ranges (presmolt and smolt in freshwater and postsmolt and adult in seawater).

Water Type	Age Range	Atlantic Salmon		Coho Salmon		Rainbow Trout		Total of Fish	Total (%)	Total of Farms	Total (%)
		Number of Healthy Fish	Number of Farms	Number of Healthy Fish	Number of Farms	Number of Healthy Fish	Number of Farms				
Freshwater	Presmolt (<50 g)	230	9			202	5	432	14.1%	14	17.9%
	Smolt (50 to 150 g)	380	9	259	7	120	4	759	24.8%	20	25.6%
Seawater	Postsmolt (150 to 800 g)	560	12	150	5	242	6	952	31.1%	23	29.5%
	Adult (>800 g)	380	8	306	7	230	6	916	29.9%	21	26.9%
Total (N)		1550	38	715	19	794	21	3059	100.0%	78	100.0%
Total (%)		50.7%	48.7%	23.4%	24.4%	26.0%	26.9%	100.0%		100.0%	

**Table 2 biology-11-01066-t002:** Reference intervals (RIs) for erythrogram biomarkers in presmolt and smolt (freshwater) and postsmolt and adult (seawater) of Atlantic salmon, coho salmon, and rainbow trout reared in Chile. The respective confidence intervals (CIs) for the respective RIs are included. Letters indicate significant differences between age ranges (*p* < 0.05).

Parameter	Abbreviation	Unit of Measure	Species	Age Stage	Median	Mean	S. D	S. E	Difference between Means	95% Reference Interval	Confidence Interval (95%)	Test	Data Processing
Hematocrit	Htc	%	Coho salmon	Smolt	53.00	53.20	7.46	0.97	a	38.31–68.49	35.58–71.19	ANOVA, Tukey	Normal
				Postsmolt	51.00	51.17	9.94	0.95	a	31.14–70.79	28.65–73.38	ANOVA, Tukey	Normal
				Adult	55.00	53.42	10.21	1.83	a	32.43–75.49	26.16–79.85	ANOVA, Tukey	Normal
			Atlantic salmon	Presmolt	37.50	40.46	7.02	1.43	b	22.41–54.60	17.27–60.64	ANOVA, Tukey	Normal
				Smolt	39.00	41.23	7.43	1.46	b	23.97–56.60	19.80–62.83	ANOVA, Tukey	Normal
				Postsmolt	47.00	46.59	6.99	0.91	a	32.62–60.94	29.72–63.28	ANOVA, Tukey	Normal
				Adult	40.00	41.24	5.25	1.27	b	29.15–52.80	25.27–57.75	ANOVA, Tukey	Normal
			Rainbow trout	Presmolt	46.00	46.83	4.39	0.80	a	36.76–55.31	34.05–58.02	ANOVA, Tukey	Normal
				Smolt	47.00	47.42	6.90	0.73	a	33.04–60.67	30.57–63.30	ANOVA, Tukey	Normal
				Postsmolt	49.00	48.72	7.27	1.29	a	34.25–64.35	29.65–69.06	ANOVA, Tukey	Normal
Red Blood Cell Count	RBC	10^7^/μL	Coho salmon	Smolt	9832.40	9389.25	3388.90	441.19	c	2225.89–16,017.98	962.34–17,318.70	ANOVA, Tukey	log(x + 1)
				Postsmolt	10,762.60	11,354.00	4526.60	414.95	b	1641.65–19,781.52	1012.74–21,512.40	ANOVA, Tukey	log(x + 1)
				Adult	15,384.60	16,389.70	6885.50	1236.70	a	4995.0–32,447.52	4995.0–32,447.52	ANOVA, Tukey	log(x + 1)
			Atlantic salmon	Presmolt	3629.70	4479.40	1663.50	339.56	a	2419.8–8436.0	2419.80–8436.0	ANOVA, Tukey	log(x + 1)
				Smolt	6371.40	6840.67	3620.80	710.11	a	2131.2–13,066.92	131.20–13,066.92	ANOVA, Tukey	log(x + 1)
				Postsmolt	5423.50	6344.49	3309.90	434.61	a	1968.25–16,083.4	1740.48–16,643.34	ANOVA, Tukey	log(x + 1)
				Adult	1953.60	2181.08	1543.90	374.44	b	319.68–6233.76	319.68–6233.76	ANOVA, Tukey	log(x + 1)
			Rainbow trout	Presmolt	3747.40	4620.12	2412.60	440.48	a	1678.32–10,966.80	1678.32–10,966.8	ANOVA, Tukey	log(x + 1)
				Smolt	4955.00	5078.64	1900.10	201.41	ab	1071.69–8683.46	508.05–9387.03	ANOVA, Tukey	log(x + 1)
				Postsmolt	3636.40	3768.45	1483.80	262.30	b	952.38–7965.36	952.38–7965.36	ANOVA, Tukey	log(x + 1)
Hemoglobin	Hgb	g/L	Coho salmon	Smolt	509.30	525.51	110.50	14.39	b	299.15–749.86	254.73–801.51	ANOVA, Tukey	Normal
				Postsmolt	607.50	599.16	107.07	10.26	a	391.02–819.18	358.59–843.14	ANOVA, Tukey	Normal
				Adult	640.80	628.51	87.76	15.76	a	454.66–823.04	394.08–861.15	ANOVA, Tukey	Normal
			Atlantic salmon	Presmolt	430.90	477.24	132.27	27.00	bc	144.71–738.62	56.0–844.63	ANOVA, Tukey	log(x + 1)
				Smolt	383.60	409.10	111.94	21.95	c	154.32–636.29	84.93–723.91	ANOVA, Tukey	log(x + 1)
				Postsmolt	545.50	566.49	110.73	14.42	a	326.56–780.92	268.60–844.32	ANOVA, Tukey	log(x + 1)
				Adult	555.60	545.63	62.70	15.21	ab	407.50–687.30	355.78–728.24	ANOVA, Tukey	log(x + 1)
			Rainbow trout	Presmolt	483.90	480.50	55.60	10.15	b	395.16–570.93	361.38–599.16	ANOVA, Tukey	BOX COX
				Smolt	520.30	548.31	154.52	16.29	a	213.48–836.16	78.77–976.65	ANOVA, Tukey	BOX COX
				Postsmolt	579.00	595.76	110.84	19.91	a	344.94–805.38	239.48–909.49	ANOVA, Tukey	BOX COX
Mean Corpuscular Volume	MCV	fL	Coho salmon	Smolt	110.9	119.74	33.12	4.31	b	40.12–177.76	22.08–196.44	ANOVA, Tukey	log(x + 1)
				Postsmolt	145.70	146.24	33.77	3.23	a	77.89–212.55	69.63–222.13	ANOVA, Tukey	log(x + 1)
				Adult	133.30	131.62	22.40	4.02	ab	84.83–178.64	73.21–189.43	ANOVA, Tukey	log(x + 1)
			Atlantic salmon	Presmolt	165.90	169.84	24.46	4.99	a	115.81–220.94	102.93–240.33	ANOVA, Tukey	log(x + 1)
				Smolt	158.60	156.26	22.58	4.43	ab	110.72–205.97	96.41–220.43	ANOVA, Tukey	log(x + 1)
				Postsmolt	138.10	145.97	24.91	3.24	b	88.55–194.02	78.10–207.21	ANOVA, Tukey	log(x + 1)
				Adult	155.60	159.71	18.37	4.45	ab	116.36–198.56	102.10–216.07	ANOVA, Tukey	log(x + 1)
			Rainbow trout	Presmolt	118.10	119.58	15.27	2.79	c	86.37–149.92	75.23–160.49	ANOVA, Tukey	log(x + 1)
				Smolt	137.20	140.97	26.41	2.78	b	86.01–192.72	79.36–202.20	ANOVA, Tukey	log(x + 1)
				Postsmolt	158.50	158.52	16.68	2.95	a	122.4–191.51	112.81–201.41	ANOVA, Tukey	log(x + 1)
Mean Corpuscular Hemoglobin Concentration	MCHC	g/L	Coho salmon	Smolt	1.00	1.02	0.17	0.02	a	0.69–1.36	0.62–1.42	ANOVA, Tukey	Normal
				Postsmolt	0.90	0.87	0.14	0.01	b	0.59–1.13	0.55–1.18	ANOVA, Tukey	Normal
				Adult	0.90	0.86	0.16	0.03	b	0.53–1.19	0.45–1.28	ANOVA, Tukey	Normal
			Atlantic salmon	Presmolt	0.90	0.89	0.22	0.05	b	0.42–1.36	0.28–1.50	ANOVA, Tukey	Normal
				Smolt	1.00	1.04	0.18	0.03	a	0.68–1.42	0.56–1.49	ANOVA, Tukey	Normal
				Postsmolt	0.80	0.84	0.15	0.02	b	0.52–1.14	0.46–1.21	ANOVA, Tukey	Normal
				Adult	0.80	0.76	0.10	0.03	b	0.52–0.98	0.44–1.08	ANOVA, Tukey	Normal
			Rainbow trout	Presmolt	0.90	0.99	0.19	0.04	a	0.68–1.15	0.60–1.23	ANOVA, Tukey	log(x + 1)
				Smolt	0.90	0.89	0.15	0.02	b	0.65–1.13	0.61–1.17	ANOVA, Tukey	log(x + 1)
				Postsmolt	0.80	0.81	0.16	0.03	b	0.62–1.06	0.55–1.13	ANOVA, Tukey	log(x + 1)

**Table 3 biology-11-01066-t003:** Reference intervals (RIs) for leukogram biomarkers in presmolt and smolt (freshwater) and postsmolt and adult (seawater) of Atlantic salmon, coho salmon and rainbow trout reared in Chile. The respective confidence intervals (CIs) for the respective RIs are included. Letters indicate significant differences between age ranges (*p* < 0.05).

Parameter	Abbreviation	Unit of Measure	Species	Age Stage	Median	Mean	S. D	S. E	Difference between Means	95% Reference Interval	Confidence Interval (95%)	Test	Data Processing
White Blood Cell Count	WBC	N°/μL	Coho salmon	Smolt	13,906.08	14,026.11	5075.93	660.83	ab	3662.20–24,199.37	2007.55–25,976.80	ANOVA, Tukey	Normal
				Postsmolt	14,385.60	14,517.42	4396.45	403.02	ab	5820.81–23,316.12	4626.99–24,548.04	ANOVA, Tukey	Normal
				Adult	12,387.60	11,976.18	3658.25	657.04	b	4325.58–19,625.58	2457.66–21,488.70	ANOVA, Tukey	Normal
			Atlantic salmon	Presmolt	15,631.02	14,927.09	3543.82	723.38	a	7426.32–22,727.96	5492.03–24,359.88	ANOVA, Tukey	log(x + 1)
				Smolt	8531.46	9510.48	3604.02	706.81	b	4528.80–16,472.40	4528.80–16,472.40	ANOVA, Tukey	log(x + 1)
				Postsmolt	12,170.04	12,797.27	3794.08	498.19	a	4816.03–20,387.69	3378.23–21,844.69	ANOVA, Tukey	log(x + 1)
				Adult	7192.80	8670.80	4206.19	1020.15	b	3247.86–18,115.20	3247.86–18,115.20	ANOVA, Tukey	log(x + 1)
			Rainbow trout	Presmolt	15,717.60	15,130.48	6015.99	1098.36	a	6493.50–29,989.98	6493.50–29,989.98	ANOVA, Tukey	Normal
				Smolt	15,984.0	15,834.26	4328.83	458.85	a	7172.65–24,508.47	5998.39–25,667.40	ANOVA, Tukey	Normal
				Postsmolt	14,635.35	15,087.67	4069.13	719.33	a	10,657.80–18,645.87	9740.24–19,738.03	ANOVA, Tukey	Normal
Lymphocytes	LYM	N°/μL	Coho salmon	Smolt	10,842.48	11,034.92	4160.62	541.67	a	2417.39–19,259.23	1149.24–20,820.34	ANOVA, Tukey	Normal
				Postsmolt	11,487.17	11,272.84	3470.47	318.14	a	4472.30–18,300.84	3498.96–19,228.08	ANOVA, Tukey	Normal
				Adult	2511.93	2543.75	1176.67	211.34	b	426.20–5388.60	426.20–5388.60	ANOVA, Tukey	Normal
			Atlantic salmon	Presmolt	11,979.79	12,302.29	2736.12	583.34	a	6543.81–18,283.54	4835.86–19,814.92	ANOVA, Tukey	Normal
				Smolt	6852.27	7427.00	2756.76	540.64	c	3849.50–13,013.20	3849.50–13,013.20	ANOVA, Tukey	Normal
				Postsmolt	9131.13	9394.85	3050.20	404.01	b	3076.28–15,484.43	2041.53–16,610.05	ANOVA, Tukey	Normal
				Adult	5665.26	6035.71	2542.45	616.63	c	2435.90–12,774.40	2435.90–12,774.40	ANOVA, Tukey	Normal
			Rainbow trout	Presmolt	13,661.82	12,758.34	4590.63	838.13	a	9704.06–17,824.74	8145.22–19,054.68	ANOVA, Tukey	BOX COX
				Smolt	12,221.97	12,004.36	1903.40	200.64	b	9113.07–15,677.21	8434.18–16,138.72	ANOVA, Tukey	BOX COX
				Postsmolt	11,489.03	11,889.42	3299.63	623.57	b	8021.64–15,157.85	7146.13–16,124.68	ANOVA, Tukey	BOX COX
Neutrophils	NEU	N°/μL	Coho salmon	Smolt	2850.48	2924.59	1180.06	153.63	b	512.42–5301.31	181.04–5723.28	ANOVA, Tukey	POWER BOX COX
				Postsmolt	2815.28	3023.69	1217.21	112.05	b	442.96–5350.66	32.73–5766.80	ANOVA, Tukey	POWER BOX COX
				Adult	9292.92	9214.03	3073.02	551.93	a	2618.39–15,383.08	1111.62–17,089.10	ANOVA, Tukey	POWER BOX COX
			Atlantic salmon	Presmolt	3201.53	3039.47	930.72	198.43	a	1071.70–5108.78	554.10–5569.84	ANOVA, Tukey	BOX COX
				Smolt	1657.21	1844.93	699.71	137.22	b	679.32–3459.20	679.32–3459.20	ANOVA, Tukey	BOX COX
				Postsmolt	3020.98	3366.32	1100.68	145.79	a	1612.89–6397.74	1551.51–6764.83	ANOVA, Tukey	BOX COX
				Adult	1948.05	2068.82	1057.98	256.60	b	772.03–4967.83	772.03–4967.83	ANOVA, Tukey	BOX COX
			Rainbow trout	Presmolt	1627.37	1781.97	1047.13	191.18	b	417.72–4523.81	417.72–4523.81	ANOVA, Tukey	BOX COX
				Smolt	1914.28	2215.56	1438.91	154.27	b	323.96–5860.65	294.51–6184.61	ANOVA, Tukey	BOX COX
				Postsmolt	2826.50	3028.72	1364.58	257.88	a	956.04–7970.69	956.04–7970.69	ANOVA, Tukey	BOX COX
Monocytes	MON	N°/μL	Coho salmon	Smolt	174.58	280.68	247.41	66.12	a	59.19–978.22	59.19–978.22	ANOVA, Tukey	log(x + 1)
				Postsmolt	251.75	304.65	187.17	23.96	a	84.95–834.96	78.59–991.01	ANOVA, Tukey	log(x + 1)
				Adult	245.75	307.76	204.13	43.52	a	83.12–796.54	83.12–796.54	ANOVA, Tukey	log(x + 1)
			Atlantic salmon	Presmolt	350.76	351.87	148.66	49.55	a	163.61–600.07	163.61–600.07	ANOVA, Tukey	log(x + 1)
				Smolt	144.52	182.02	123.89	34.36	b	51.77–445.24	51.77–445.24	ANOVA, Tukey	log(x + 1)
				Postsmolt	256.41	258.11	110.56	24.13	ab	86.58–496.30	86.58–496.30	ANOVA, Tukey	log(x + 1)
				Smolt	84.18	162.21	173.82	86.91	b	58.50–421.98	58.50–421.98	ANOVA, Tukey	log(x + 1)
			Rainbow trout	Postsmolt	139.24	188.44	145.54	48.51	b	65.36–447.55	65.36–447.55	Kruskal-Wallis, Dunn	
				Adult	294.51	446.15	387.88	47.39	a	147.25–1472.53	250.05–1472.53	Kruskal-Wallis, Dunn	
				Presmolt	208.86	269.41	190.49	54.99	ab	121.57–703.30	121.57–703.30	Kruskal-Wallis, Dunn	
Thrombocyte count	TCC	N°/μL	Coho salmon	Smolt	5688.88	5550.22	1898.31	247.14	a	1632.27–9319.54	930.44–10,048.76	ANOVA, Tukey	BOX COX
				Postsmolt	5967.36	6003.87	1623.96	148.87	a	2705.02–9166.50	2275.34–9644.43	ANOVA, Tukey	BOX COX
				Adult	6433.56	6252.88	1364.83	245.13	a	3479.77–9212.12	2739.57–9871.98	ANOVA, Tukey	BOX COX
			Atlantic salmon	Presmolt	3184.88	3468.78	1386.72	283.06	a	1606.39–6600.73	1606.39–6600.73	ANOVA, Tukey	Normal
				Smolt	3491.17	3854.07	1852.63	363.33	a	1594.14–7948.71	1594.14–7948.71	ANOVA, Tukey	Normal
				Postsmolt	3562.63	4173.17	1757.30	234.83	a	1763,70–8826.51	1759.04–9224.77	ANOVA, Tukey	Normal
				Adult	1566.52	1660.49	963.82	233.76	b	357.26–3985.34	357.26–3985.34	ANOVA, Tukey	Normal
			Rainbow trout	Postsmolt	3449.88	3533.37	1629.58	302.61	a	1438.56–7384.61	1438.56–7384.61	ANOVA, Tukey	log(x + 1)
				Adult	3866.66	3779.87	1198.54	127.05	ab	1308.01–6113.98	961.13–6498.79	ANOVA, Tukey	log(x + 1)
				Presmolt	2781.35	2935.75	971.37	177.35	b	797.91–4886.41	213.08–5488.10	ANOVA, Tukey	log(x + 1)

**Table 4 biology-11-01066-t004:** Reference intervals (RIs) for plasma substrates biomarkers in presmolt, smolt, postsmolt and adult of Atlantic salmon, coho salmon and rainbow trout reared in Chile. The respective confidence intervals (CIs) for the respective RIs are included. Letters indicate significant differences between age ranges (*p* < 0.05).

Parameter	Abbreviation	Unit of Measure	Species	Age Stage	Median	Mean	S. D	S. E	Difference between Means	95% Reference Interval	Confidence Interval (95%)	Test	Data Processing
Total protein	TPO	g/L	Coho salmon	Smolt	40.00	39.33	10.00	0.64	b	20.03–59.61	18.21–61.17	ANOVA, Tukey	Normal
				Postsmolt	43.90	43.42	6.46	0.60	a	30.69–56.50	28.88–58.06	ANOVA, Tukey	Normal
				Adult	41.20	42.24	7.63	0.41	a	26.58–56.92	25.32–58.30	ANOVA, Tukey	Normal
			Atlantic salmon	Presmolt	43.80	44.97	8.61	0.64	b	27.38–61.79	25.34–63.69	ANOVA, Tukey	Normal
				Smolt	45.50	45.39	9.96	0.52	b	25.67–64.90	24.35–66.29	ANOVA, Tukey	Normal
				Postsmolt	44.50	44.60	7.61	0.33	a	29.60–59.54	28.67–60.44	ANOVA, Tukey	Normal
				Adult	49.80	48.29	7.90	0.41	b	33.75–65.39	32.27–66.60	ANOVA, Tukey	Normal
			Rainbow trout	Presmolt	33.00	34.42	10.09	1.84	c	13.05–55.80	7.44–63.16	ANOVA, Tukey	Normal
				Smolt	41.00	40.15	7.63	0.45	b	25.11–55.38	23.86–56.61	ANOVA, Tukey	Normal
				Postsmolt	41.00	41.53	6.56	0.42	a	28.17–54.17	27.06–55.39	ANOVA, Tukey	Normal
				Adult	46.90	46.52	8.93	0.60	b	28.79–64.13	27.06–65.80	ANOVA, Tukey	Normal
Albumins	ALB	g/L	Coho salmon	Smolt	15.70	15.77	2.73	0.18	b	10.21–21.0	9.70–21.56	ANOVA, Tukey	log(x + 1)
				Postsmolt	16.60	16.73	2.75	0.26	a	11.28–22.26	10.58–22.91	ANOVA, Tukey	log(x + 1)
				Adult	16.20	16.35	2.38	0.13	a	11.40–20.80	11.00–21.30	ANOVA, Tukey	log(x + 1)
			Atlantic salmon	Presmolt	16.60	16.81	2.26	0.17	b	12.31–21.32	11.87–21.75	ANOVA, Tukey	Normal
				Smolt	18.15	18.23	3.29	0.18	a	11.66–24.65	11.21–25.16	ANOVA, Tukey	Normal
				Postsmolt	17.90	18.03	2.44	0.11	a	13.09–22.70	12.77–23.03	ANOVA, Tukey	Normal
				Adult	18.60	18.47	2.67	0.14	a	13.45–23.97	13.01–24.33	ANOVA, Tukey	Normal
			Rainbow trout	Presmolt	15.45	15.49	2.87	0.17	c	9.81–21.11	9.34–21.60	ANOVA, Tukey	Normal
				Smolt	16.50	16.49	2.28	0.15	b	11.91–20.91	11.50–21.35	ANOVA, Tukey	Normal
				Postsmolt	18.40	18.62	3.43	0.23	a	11.68–25.27	11.08–25.96	ANOVA, Tukey	Normal
Globulins	GLO	g/L	Coho salmon	Smolt	26.20	25.61	7.00	0.46	a	11.83–39.57	10.61–40.69	ANOVA, Tukey	Normal
				Postsmolt	26.70	26.68	4.43	0.41	a	17.71–35.36	16.64–36.54	ANOVA, Tukey	Normal
				Adult	25.50	25.91	6.08	0.33	a	13.60–37.60	12.70–38.59	ANOVA, Tukey	Normal
			Atlantic salmon	Presmolt	27.20	27.74	6.50	0.49	b	14.59–40.42	13.40–41.79	Kruskal-Wallis, Dunn	
				Smolt	27.90	27.31	8.53	0.47	b	10.88–44.06	9.53–45.06	Kruskal-Wallis, Dunn	
				Postsmolt	26.50	26.64	6.32	0.28	b	14.37–39.11	13.58–39.88	Kruskal-Wallis, Dunn	
				Adult	30.75	29.83	6.02	0.31	a	18.78–42.34	17.75–43.21	Kruskal-Wallis, Dunn	
			Rainbow trout	Smolt	25.00	24.77	5.56	0.33	b	13.71–35.68	12.80–36.59	ANOVA, Tukey	Normal
				Postsmolt	24.60	25.04	5.58	0.36	b	13.67–35.77	12.69–36.91	ANOVA, Tukey	Normal
				Adult	28.00	27.95	6.61	0.45	a	14.95–41.06	13.74–42.24	ANOVA, Tukey	Normal
Total bilirubin	TBI	μmol/L	Coho salmon	Smolt	2.48	2.79	0.55	0.08	b	1.7–3.87	1.60–3.97	Kruskal-Wallis, Dunn	
				Adult	3.55	3.58	0.60	0.15	a	2.23–4.90	1.79–5.39	Kruskal-Wallis, Dunn	
			Atlantic salmon	Presmolt	2.90	2.97	0.12	0.07	a	2.74–3.19	2.72–3.22	ANOVA, Tukey	POWER BOX COX
				Smolt	2.90	2.96	0.32	0.06	a	2.22–3.58	1.90–3.88	ANOVA, Tukey	POWER BOX COX
				Postsmolt	2.95	3.03	0.40	0.08	a	2.16–3.88	1.93–4.08	ANOVA, Tukey	POWER BOX COX
				Adult	2.80	3.06	0.72	0.18	a	1.11–4.37	0.19–5.46	ANOVA, Tukey	POWER BOX COX
			Rainbow trout	Smolt	3.00	3.18	0.58	0.16	a	2.11–3.92	1.76–4.33	ANOVA, Tukey	POWER BOX COX
				Postsmolt	3.20	3.30	0.17	0.10	a	2.96–3.64	2.93–3.67	ANOVA, Tukey	POWER BOX COX
				Adult	2.90	3.06	0.34	0.11	a	2.09–3.95	1.71–4.41	ANOVA, Tukey	POWER BOX COX
Direct bilirubin	DBI	μmol/L	Coho salmon	Smolt	1.88	2.05	0.40	0.10	a	1.03–2.95	0.82–3.23	ANOVA, Tukey	BOX COX
				Adult	1.90	1.98	0.40	0.07	a	1.10–2.80	0.78–3.04	ANOVA, Tukey	BOX COX
			Atlantic salmon	Presmolt	1.80	1.83	0.26	0.13	ab	1.31–2.34	1.26–2.39	ANOVA, Tukey	POWER BOX COX
				Smolt	1.80	1.88	0.32	0.05	b	1.31–2.29	1.20–2.40	ANOVA, Tukey	POWER BOX COX
				Postsmolt	2.10	2.16	0.36	0.06	a	1.47–2.76	1.31–2.90	ANOVA, Tukey	POWER BOX COX
				Adult	1.80	1.87	0.27	0.05	b	1.26–2.43	1.12–2.60	ANOVA, Tukey	POWER BOX COX
			Rainbow trout	Smolt	1.70	1.78	0.28	0.06	a	1.09–2.29	0.89–2.56	ANOVA, Tukey	POWER BOX COX
				Postsmolt	1.60	1.77	0.29	0.17	a	1.20–2.33	1.15–2.38	ANOVA, Tukey	POWER BOX COX
				Adult	1.60	1.57	0.09	0.03	a	1.39–1.75	1.37–1.77	ANOVA, Tukey	POWER BOX COX
Creatinine	CRE	μmol/L	Coho salmon	Smolt	24.64	26.67	9.15	1.08	a	6.63–44.29	3.61–48.26	Kruskal-Wallis, Dunn	
				Postsmolt	23.87	21.51	4.87	2.81	a	15.91–24.75	15.91–24.75	Kruskal-Wallis, Dunn	
				Adult	22.10	24.21	8.04	1.02	a	5.56–39.03	2.30–43.24	Kruskal-Wallis, Dunn	
			Atlantic salmon	Presmolt	23.90	26.91	10.27	1.00	b	6.79–47.04	4.78–49.05	ANOVA, Tukey	log(x + 1)
				Smolt	25.64	26.72	7.67	0.51	b	11.09–41.70	9.67–43.21	ANOVA, Tukey	log(x + 1)
				Postsmolt	31.90	31.28	8.80	0.85	a	13.94–49.13	11.50–51.25	ANOVA, Tukey	log(x + 1)
				Adult	23.90	24.32	5.57	0.92	b	11.90–34.83	9.09–38.17	ANOVA, Tukey	log(x + 1)
			Rainbow trout	Presmolt	61.44	59.63	12.95	2.90	a	33.25–89.47	22.86–101.49	ANOVA, Tukey	POWER BOX COX
				Smolt	28.68	32.01	13.77	1.44	b	5.03–58.99	2.77–61.26	ANOVA, Tukey	POWER BOX COX
				Postsmolt	26.52	27.77	8.18	0.99	b	10.60–43.93	8.00–46.86	ANOVA, Tukey	POWER BOX COX
				Adult	26.08	27.35	8.74	1.15	b	8.60–44.26	4.00–48.23	ANOVA, Tukey	POWER BOX COX
Glucose	GLU	mmol/L	Coho salmon	Smolt	4.29	4.65	1.51	0.13	a	1.34–7.51	0.94–7.98	ANOVA, Tukey	log(x + 1)
				Postsmolt	4.58	4.61	0.76	0.07	a	3.11–6.12	2.92–6.31	ANOVA, Tukey	log(x + 1)
				Adult	2.85	3.07	1.47	0.15	b	1.10–5.96	1.02–5.99	ANOVA, Tukey	log(x + 1)
			Atlantic salmon	Presmolt	5.40	5.40	1.85	0.13	b	1.77–9.02	1.40–9.39	Kruskal-Wallis, Dunn	
				Smolt	6.10	6.33	1.68	0.09	a	2.91–9.60	2.68–9.87	Kruskal-Wallis, Dunn	
				Postsmolt	4.10	4.06	1.65	0.08	c	0.76–7.06	0.57–7.29	Kruskal-Wallis, Dunn	
				Adult	5.50	5.70	0.81	0.05	b	4.0–7.30	3.86–7.43	Kruskal-Wallis, Dunn	
			Rainbow trout	Presmolt	2.79	2.76	0.56	0.10	c	1.60–3.93	1.36–4.16	Kruskal-Wallis, Dunn	
				Smolt	2.76	3.13	1.82	0.12	c	1.08–7.48	1.06–8.46	Kruskal-Wallis, Dunn	
				Postsmolt	4.08	4.32	2.13	0.18	b	0.15–8.49	8.11–8.88	Kruskal-Wallis, Dunn	
				Adult	5.24	5.04	2.05	0.17	a	0.92–9.12	0.51–9.60	Kruskal-Wallis, Dunn	
Lactate	LAC	mmol/L	Coho salmon	Smolt	4.69	5.44	3.20	0.23	b	1.25–13.17	1.12–13.81	Kruskal-Wallis, Dunn	
				Postsmolt	5.15	5.47	1.78	0.16	a	1.67–8.84	1.21–9.43	Kruskal-Wallis, Dunn	
				Adult	5.63	5.81	1.85	0.11	b	1.90–9.22	1.59–9.59	Kruskal-Wallis, Dunn	
			Atlantic salmon	Presmolt	4.98	5.60	2.21	0.17	a	3.03–11.44	3.0–12.59	Kruskal-Wallis, Dunn	
				Smolt	5.10	5.77	2.16	0.13	a	3.10–11.29	3.0–13.18	Kruskal-Wallis, Dunn	
				Postsmolt	4.80	5.25	2.33	0.11	b	1.56–11.35	1.40–11.74	Kruskal-Wallis, Dunn	
				Adult	3.80	3.84	0.94	0.06	a	1.97–5.69	1.82–5.85	Kruskal-Wallis, Dunn	
			Rainbow trout	Presmolt	11.77	12.27	2.84	0.52	c	8.05–19.26	8.05–19.26	Kruskal-Wallis, Dunn	
				Smolt	6.55	7.00	3.54	0.22	c	2.27–14.18	2.0–14.97	Kruskal-Wallis, Dunn	
				Postsmolt	6.00	6.23	2.38	0.15	b	1.38–10.84	0.96–11.29	Kruskal-Wallis, Dunn	
				Adult	4.72	4.88	1.99	0.14	a	0.77–8.65	0.41–9.11	Kruskal-Wallis, Dunn	
Urea	URE	mmol/L	Coho salmon	Smolt	1.80	1.76	0.27	0.03	a	1.21–2.30	1.13–2.39	ANOVA, Tukey	Normal
				Postsmolt	0.90	0.91	0.20	0.02	b	0.52–1.31	0.45–1.39	ANOVA, Tukey	Normal
				Adult	1.60	1.60	0.35	0.06	c	0.89–2.33	0.70–2.51	ANOVA, Tukey	Normal
			Atlantic salmon	Presmolt	1.45	1.40	0.26	0.03	a	0.87–1.94	0.78–2.03	ANOVA, Tukey	BOX COX
				Smolt	1.10	1.14	0.30	0.04	b	0.51–1.76	0.37–1.88	ANOVA, Tukey	BOX COX
				Postsmolt	1.50	1.50	0.19	0.03	a	1.09–1.86	0.97–1.98	ANOVA, Tukey	BOX COX
				Adult	1.10	1.09	0.30	0.05	b	0.46–1.70	0.32–1.86	ANOVA, Tukey	BOX COX
			Rainbow trout	Presmolt	1.20	1.20	0.09	0.02	a	1.0–1.38	0.94–1.45	ANOVA, Tukey	BOX COX
				Smolt	1.00	1.07	0.19	0.02	b	0.65–1.42	0.59–1.48	ANOVA, Tukey	BOX COX
				Postsmolt	1.30	1.27	0.16	0.03	a	0.92–1.58	0.83–1.69	ANOVA, Tukey	BOX COX
Uric acid	UAC	μmol/L	Coho salmon	Smolt	1.80	1.76	0.27	0.03	a	1.21–2.30	1.12–2.39	ANOVA, Tukey	Normal
				Postsmolt	0.90	0.91	0.20	0.02	c	0.52–1.31	0.45–1.38	ANOVA, Tukey	Normal
				Adult	1.60	1.60	0.35	0.06	b	0.89–2.33	0.71–2.51	ANOVA, Tukey	Normal
			Atlantic salmon	Presmolt	1.45	1.40	0.26	0.03	a	0.87–1.94	0.78–2.33	ANOVA, Tukey	log(x + 1)
				Smolt	1.10	1.13	0.28	0.04	b	0.55–1.70	0.44–1.8	ANOVA, Tukey	log(x + 1)
				Postsmolt	1.50	1.49	0.16	0.02	a	1.15–1.80	1.08–1.87	ANOVA, Tukey	log(x + 1)
				Adult	1.10	1.09	0.30	0.05	b	0.46–1.70	0.32–1.86	ANOVA, Tukey	log(x + 1)
			Rainbow trout	Presmolt	1.20	1.20	0.09	0.02	a	1.0–1.38	0.94–1.46	Kruskal-Wallis, Dunn	
				Smolt	1.00	1.06	0.17	0.02	b	0.68–1.39	0.63–1.44	Kruskal-Wallis, Dunn	
				Postsmolt	1.20	1.24	0.12	0.02	a	0.97–1.51	0.93–1.56	Kruskal-Wallis, Dunn	
Ammonia	NH_3_	mmol/L	Coho salmon	Smolt	2.92	2.75	0.60	0.08	a	1.58–4.09	1.29–4.32	Kruskal-Wallis, Dunn	
				Postsmolt	1.07	0.98	0.34	0.03	b	0.36–1.76	0.24–1.83	Kruskal-Wallis, Dunn	
				Adult	2.62	2.51	0.50	0.10	a	1.53–3.66	1.18–4.06	Kruskal-Wallis, Dunn	
			Atlantic salmon	Presmolt	1.92	1.83	0.34	0.05	b	1.22–2.63	1.02–2.75	Kruskal-Wallis, Dunn	
				Smolt	1.85	1.51	0.68	0.09	b	0.19–2.84	0.09–2.94	Kruskal-Wallis, Dunn	
				Postsmolt	1.83	1.88	0.22	0.03	b	1.38–2.30	1.25–2.44	Kruskal-Wallis, Dunn	
				Adult	1.24	1.13	0.36	0.07	a	0.36–1.95	0.16–2.11	Kruskal-Wallis, Dunn	
			Rainbow trout	Presmolt	1.71	1.73	0.13	0.02	a	1.46–1.99	1.40–2.05	Kruskal-Wallis, Dunn	
				Smolt	1.63	1.62	0.32	0.03	a	0.97–2.27	0.83–2.41	Kruskal-Wallis, Dunn	
				Postsmolt	1.69	1.72	0.24	0.04	a	1.21–2.20	1.10–2.34	Kruskal-Wallis, Dunn	
Total Cholesterol	TCH	mmol/L	Coho salmon	Smolt	7.14	7.76	2.95	0.19	c	1.41–13.30	0.99–13.94	Kruskal-Wallis, Dunn	
				Postsmolt	9.04	8.86	1.67	0.16	a	5.62–12.29	5.11–12.77	Kruskal-Wallis, Dunn	
				Adult	8.09	8.13	2.84	0.16	b	2.46–13.65	1.96–14.11	Kruskal-Wallis, Dunn	
			Atlantic salmon	Presmolt	12.32	12.78	2.74	0.20	a	7.17–18.17	6.67–18.72	Kruskal-Wallis, Dunn	
				Smolt	11.10	11.44	3.66	0.20	b	4.08–18.56	3.62–19.09	Kruskal-Wallis, Dunn	
				Postsmolt	7.50	7.69	2.63	0.11	d	2.39–12.78	2.05–13.11	Kruskal-Wallis, Dunn	
				Adult	8.80	9.06	1.90	0.10	c	5.05–12.59	4.69–12.96	Kruskal-Wallis, Dunn	
			Rainbow trout	Presmolt	5.34	6.08	2.44	0.45	b	2.07–12.07	2.07–12.07	ANOVA, Tukey	POWER BOX COX
				Smolt	8.94	9.08	3.19	0.19	a	2.54–15.14	1.96–15.73	ANOVA, Tukey	POWER BOX COX
				Postsmolt	6.78	6.87	2.56	0.17	b	1.65–11.79	1.21–12.29	ANOVA, Tukey	POWER BOX COX
				Adult	5.91	6.34	2.85	0.20	b	0.74–11.93	0.21–12.47	ANOVA, Tukey	POWER BOX COX
Triglycerides	TRG	mmol/L	Coho salmon	Smolt	2.84	2.90	1.31	0.09	a	0.34–5.47	0.11–5.70	ANOVA, Tukey	POWER BOX COX
				Postsmolt	2.73	3.12	1.26	0.12	a	1.57–6.94	1.48–7.58	ANOVA, Tukey	POWER BOX COX
				Adult	2.77	2.91	1.24	0.07	a	0.34–5.24	0.13–5.45	ANOVA, Tukey	POWER BOX COX
			Atlantic salmon	Presmolt	4.25	4.54	1.42	0.11	c	1.4–7.15	0.98–7.60	Kruskal-Wallis, Dunn	
				Smolt	4.10	4.19	1.71	0.09	a	0.54–7.29	0.29–7.60	Kruskal-Wallis, Dunn	
				Postsmolt	2.80	3.01	1.39	0.06	b	0.27–5.74	0.10–5.91	Kruskal-Wallis, Dunn	
				Adult	2.75	3.20	1.59	0.09	b	0.31–6.19	0.60–6.46	Kruskal-Wallis, Dunn	
			Rainbow trout	Presmolt	6.73	7.40	2.62	0.50	a	1.63–12.97	0.56–14.27	ANOVA, Tukey	POWER BOX COX
				Smolt	5.00	4.99	1.85	0.11	b	1.26–8.55	1.0–8.89	ANOVA, Tukey	POWER BOX COX
				Postsmolt	4.20	4.70	2.09	0.14	b	0.59–8.80	0.22–9.18	ANOVA, Tukey	POWER BOX COX
				Adult	4.33	4.73	2.11	0.15	b	0.6–8.86	0.20–9.26	ANOVA, Tukey	POWER BOX COX
High-density lipoprotein cholesterol	HDL	mmol/L	Coho salmon	Smolt	3.39	3.49	1.49	0.20	b	0.57–6.40	0.31–6.66	Kruskal-Wallis, Dunn	
				Postsmolt	5.65	5.45	1.62	0.16	a	2.26–8.76	1.82–9.20	Kruskal-Wallis, Dunn	
				Adult	0.28	0.32	0.11	0.02	c	0.10–0.54	0.09–0.56	Kruskal-Wallis, Dunn	
			Atlantic salmon	Presmolt	1.86	1.98	0.67	0.10	c	0.45–3.19	0.14–3.56	ANOVA, Tukey	Normal
				Smolt	5.41	5.60	1.27	0.17	b	2.94–8.14	2.49–8.61	ANOVA, Tukey	Normal
				Postsmolt	6.61	6.23	1.54	0.21	a	3.18–9.61	2.47–10.11	ANOVA, Tukey	Normal
				Adult	6.88	6.90	1.21	0.27	a	4.36–9.59	3.48–10.39	ANOVA, Tukey	Normal
			Rainbow trout	Presmolt	1.63	1.62	0.44	0.08	c	0.69–2.53	0.48–2.74	ANOVA, Tukey	Normal
				Smolt	4.46	4.26	1.64	0.18	b	0.99–7.61	0.54–8.04	ANOVA, Tukey	Normal
				Postsmolt	2.53	2.50	0.90	0.17	a	0.65–4.40	0.15–4.86	ANOVA, Tukey	Normal
Low-density lipoprotein cholesterol	LDL	mmol/L	Coho salmon	Smolt	1.20	1.34	0.95	0.12	c	0.15–4.25	0.30–5.30	ANOVA, Tukey	BOX COX
				Postsmolt	1.57	1.60	0.51	0.05	b	0.57–2.59	0.40–2.75	ANOVA, Tukey	BOX COX
				Adult	2.10	2.20	1.02	0.19	a	0.19–4.20	0.04–4.36	ANOVA, Tukey	BOX COX
			Atlantic salmon	Presmolt	2.35	2.52	0.90	0.16	a	0.75–4.29	0.57–4.47	ANOVA, Tukey	BOX COX
				Smolt	2.42	2.40	1.00	0.13	a	0.33–4.40	0.04–4.77	ANOVA, Tukey	BOX COX
				Postsmolt	1.55	1.62	0.56	0.07	b	0.39–2.66	0.16–2.93	ANOVA, Tukey	BOX COX
				Adult	0.90	0.97	0.33	0.06	c	0.22–1.63	0.04–1.87	ANOVA, Tukey	BOX COX
			Rainbow trout	Presmolt	1.45	1.46	0.60	0.11	b	0.50–3.10	0.50–3.10	ANOVA, Tukey	BOX COX
				Smolt	1.89	1.98	0.96	0.12	a	0.41–4.74	0.60–4.88	ANOVA, Tukey	BOX COX
				Postsmolt	2.40	2.50	1.20	0.22	a	0.36–4.79	0.94–4.79	ANOVA, Tukey	BOX COX

**Table 5 biology-11-01066-t005:** Reference intervals (RIs) for plasma enzymes biomarkers in presmolt and smolt (freshwater) and postsmolt and adult (seawater) of Atlantic salmon, coho salmon, and rainbow trout reared in Chile. The respective confidence intervals (CIs) for the respective RIs are included. Letters indicate significant differences between age ranges (*p* < 0.05).

Parameter	Abbreviation	Unit of Measure	Species	Age Stage	Median	Mean	S. D	S. E	Difference between Means	95% Reference Interval	Confidence Interval (95%)	Test	Data Processing
Alkaline phosphatase	ALP	U/L	Coho salmon	Smolt	43.00	48.74	25.86	1.85	b	18.93–115.23	15.0–122.0	Kruskal-Wallis, Dunn	
				Postsmolt	67.00	65.80	22.58	2.18	a	20.56–110.74	15.04–116.44	Kruskal-Wallis, Dunn	
				Adult	39.00	46.68	26.40	1.65	c	13.00–106.2	11.0–255.49	Kruskal-Wallis, Dunn	
			Atlantic salmon	Presmolt	177.00	179.77	35.12	2.73	a	107.55–247.14	99.80–255.49	Kruskal-Wallis, Dunn	
				Smolt	176.00	176.56	38.76	2.18	a	99.25–252.11	93.88–258.23	Kruskal-Wallis, Dunn	
				Postsmolt	156.00	153.59	41.11	2.10	b	70.50–232.67	64.90–238.67	Kruskal-Wallis, Dunn	
				Adult	171.00	170.62	34.77	2.10	a	101.98–239.18	96.24–244.93	Kruskal-Wallis, Dunn	
			Rainbow trout	Presmolt	164.00	161.00	27.71	5.54	a	101.37–218.62	84.47–234.88	Kruskal-Wallis, Dunn	
				Smolt	83.00	90.14	41.69	2.74	c	8.42–171.85	1.57–178.71	Kruskal-Wallis, Dunn	
				Postsmolt	99.00	104.74	46.80	3.09	b	7.95–194.24	0.57–204.26	Kruskal-Wallis, Dunn	
				Adult	65.50	74.77	36.68	2.53	d	22.28–157.73	20.0–162.0	Kruskal-Wallis, Dunn	
Alanine transaminase	ALT	U/L	Coho salmon	Smolt	15.90	15.86	7.41	0.59	b	1.35–30.38	0.05–31.68	ANOVA, Tukey	POWER BOX COX
				Postsmolt	10.30	10.91	3.61	0.36	a	3.13–17.73	1.98–18.91	ANOVA, Tukey	POWER BOX COX
				Adult	14.80	15.38	5.39	0.35	c	4.31–25.73	3.54–26.78	ANOVA, Tukey	POWER BOX COX
			Atlantic salmon	Presmolt	19.95	19.70	7.81	0.64	c	4.62–35.63	2.70–37.38	ANOVA, Tukey	POWER BOX COX
				Smolt	15.20	16.88	7.41	0.41	b	1.04–30.98	0.09–32.38	ANOVA, Tukey	POWER BOX COX
				Postsmolt	14.10	14.47	6.75	0.36	a	1.25–27.69	0.44–28.50	ANOVA, Tukey	POWER BOX COX
				Adult	12.65	12.73	4.68	0.30	a	3.28–21.74	2.45–22.68	ANOVA, Tukey	POWER BOX COX
			Rainbow trout	Presmolt	6.60	7.44	1.70	0.37	c	3.00–11.08	1.74–12.27	ANOVA, Tukey	POWER BOX COX
				Smolt	14.50	15.87	7.83	0.48	a	5.40–33.90	5.37–34.67	ANOVA, Tukey	POWER BOX COX
				Postsmolt	10.90	11.61	4.84	0.37	b	1.47–20.85	0.39–22.15	ANOVA, Tukey	POWER BOX COX
				Adult	8.20	10.33	4.95	0.38	d	5.50–25.79	5.14–29.30	ANOVA, Tukey	POWER BOX COX
Aspartate aminotransferase	AST	U/L	Coho salmon	Smolt	245.50	269.36	134.75	9.11	b	52.0–582.85	38.10–601.50	ANOVA, Tukey	POWER BOX COX
				Postsmolt	271.30	293.45	92.50	8.70	a	86.95–465.02	52.98–500.03	ANOVA, Tukey	POWER BOX COX
				Adult	259.30	291.12	125.28	6.91	c	45.57–536.67	26.50–555.74	ANOVA, Tukey	POWER BOX COX
			Atlantic salmon	Presmolt	346.00	353.18	92.48	7.96	c	160.02–528.15	136.88–555.85	ANOVA, Tukey	POWER BOX COX
				Smolt	352.10	367.40	106.57	6.20	b	134.57–558.92	115.90–584.67	ANOVA, Tukey	POWER BOX COX
				Postsmolt	267.10	272.41	107.76	4.99	a	49.99–474.44	34.97–490.75	ANOVA, Tukey	POWER BOX COX
				Adult	278.10	286.64	112.45	6.16	a	52.49–496.83	35.13–518.01	ANOVA, Tukey	POWER BOX COX
			Rainbow trout	Presmolt	346.65	362.46	112.69	20.57	c	110.81–584.47	51.37–653.85	ANOVA, Tukey	POWER BOX COX
				Smolt	349.20	348.92	124.85	7.80	a	99.19–591.83	73.07–615.53	ANOVA, Tukey	POWER BOX COX
				Postsmolt	276.45	290.74	110.98	7.25	b	54.44–496.13	31.02–519.56	ANOVA, Tukey	POWER BOX COX
				Adult	307.00	316.66	129.73	9.00	d	50.78–565.15	24.77–594.71	ANOVA, Tukey	POWER BOX COX
Total amylase	TAM	U/L	Coho salmon	Smolt	990.50	972.68	502.26	44.39	c	216.13–2366.88	163.50–2662.95	ANOVA, Tukey	BOX COX
				Postsmolt	1539.00	1593.84	417.81	39.30	a	711.29–2388.73	599.84–3517.48	ANOVA, Tukey	BOX COX
				Adult	1281.50	1267.71	544.66	34.18	b	143.61–2294.0	53.19–2404.72	ANOVA, Tukey	BOX COX
			Atlantic salmon	Presmolt	609.00	656.76	303.19	22.85	a	62.53–1250.99	3.15–1310.37	Kruskal-Wallis, Dunn	
				Smolt	880.00	924.00	303.60	16.61	b	306.71–1515.83	264.88–1562.42	Kruskal-Wallis, Dunn	
				Postsmolt	967.90	987.36	381.63	17.91	b	208.88–1712.55	140.99–1777.90	Kruskal-Wallis, Dunn	
				Adult	1177.00	1180.59	341.57	21.22	c	491.25–1832.17	422.24–1898.45	Kruskal-Wallis, Dunn	
			Rainbow trout	Presmolt	445.00	493.08	220.81	43.30	c	160.0–1050.0	160.0–1050.0	Kruskal-Wallis, Dunn	
				Smolt	876.00	887.59	445.30	27.78	b	155.60–1583.10	135.45–2414.80	Kruskal-Wallis, Dunn	
				Postsmolt	934.00	991.61	399.42	25.89	a	160.73–1754.25	79.31–1836.25	Kruskal-Wallis, Dunn	
				Adult	1120.00	1069.92	352.98	26.02	a	357.87–1768.88	276.34–1844.43	Kruskal-Wallis, Dunn	
Lipase	LIP	U/L	Coho salmon	Smolt	5.40	5.53	0.83	0.07	b	4.04–6.89	3.88–7.06	ANOVA, Tukey	BOX COX
				Postsmolt	5.70	5.69	0.40	0.04	ab	4.91–6.50	4.79–6.64	ANOVA, Tukey	BOX COX
				Adult	5.90	5.85	0.88	0.05	a	4.19–7.59	4.05–7.72	ANOVA, Tukey	BOX COX
			Atlantic salmon	Presmolt	6.20	6.02	0.86	0.06	a	4.35–7.83	4.05–8.12	Kruskal-Wallis, Dunn	
				Smolt	5.90	5.63	1.21	0.06	b	3.39–8.26	3.14–8.48	Kruskal-Wallis, Dunn	
				Postsmolt	5.20	5.33	1.70	0.08	c	2.01–8.46	1.81–8.68	Kruskal-Wallis, Dunn	
				Adult	5.60	5.42	1.44	0.08	c	2.83–8.12	2.59–8.34	Kruskal-Wallis, Dunn	
			Rainbow trout	Presmolt	6.60	6.50	0.57	0.13	a	5.23–7.70	4.79–8.14	Kruskal-Wallis, Dunn	
				Smolt	5.30	5.33	0.74	0.04	c	3.83–6.76	3.70–6.91	Kruskal-Wallis, Dunn	
				Postsmolt	5.60	5.62	0.81	0.05	b	3.96–7.15	3.80–7.30	Kruskal-Wallis, Dunn	
				Adult	5.60	5.47	1.34	0.09	b	2.88–8.21	2.59–8.44	Kruskal-Wallis, Dunn	
Creatine kinase total	CKT	U/L	Coho salmon	Smolt	5212.50	5980.72	4527.46	377.29	b	23.0–16,269.25	16.0–16,967.75	ANOVA, Tukey	POWER BOX COX
				Postsmolt	7926.00	8324.64	6839.67	687.41	a	22.0–23,723.00	22.0–26,210.0	ANOVA, Tukey	POWER BOX COX
				Adult	5181.00	7207.93	5716.01	340.99	c	457.0–22,267.10	293.0–23,667.0	ANOVA, Tukey	POWER BOX COX
			Atlantic salmon	Presmolt	7548.00	8670.21	4816.54	455.12	c	1398.98–20,848.55	1014.0–21,725.0	ANOVA, Tukey	POWER BOX COX
				Smolt	7029.50	8638.65	4422.61	351.84	b	2827.35–18,632.50	2522.90–21,403.50	ANOVA, Tukey	POWER BOX COX
				Postsmolt	4637.00	4787.97	1878.53	114.11	a	883.94–8317.86	552.21–8674.11	ANOVA, Tukey	POWER BOX COX
				Adult	5670.50	6144.78	3025.80	198.65	a	1921.18–14,481.05	1347.0–16,082.0	ANOVA, Tukey	POWER BOX COX
			Rainbow trout	Presmolt	2705.00	3414.33	2052.56	374.74	c	1380.0–9180.0	1380.0–9180.0	ANOVA, Tukey	POWER BOX COX
				Smolt	5294.00	6660.63	5120.90	368.61	b	265.75–20,177.30	24.0–20,617.0	ANOVA, Tukey	POWER BOX COX
				Postsmolt	5250.00	7235.51	6037.84	390.56	ab	751.95–23,438.70	421.0–23,963.0	ANOVA, Tukey	POWER BOX COX
				Adult	5965.00	7875.02	5099.61	386.60	a	1553.48–21,792.40	1273.0–22,850.0	ANOVA, Tukey	POWER BOX COX
Cardiac creatine kinase isoenzyme	CK-MB	U/L	Coho salmon	Smolt	3419.50	3628.47	1410.47	190.19	b	626.15–6386.21	45.34–6982.88	Kruskal-Wallis, Dunn	
				Postsmolt	9707.00	10,268.18	5107.02	484.74	a	1706.40–22,365.60	1248.0–23,554.0	Kruskal-Wallis, Dunn	
				Adult	3811.00	4445.89	2760.02	235.80	b	1244.57–13,392.15	1059.78–19,625.45	Kruskal-Wallis, Dunn	
			Atlantic salmon	Presmolt	11,277.00	11,012.54	6265.61	934.02	c	169.67–22,306.20	17.31–22,362.0	Kruskal-Wallis, Dunn	
				Smolt	16,970.00	16,911.02	4926.72	703.82	b	6946.41–26,975.05	4820.19–29,099.54	Kruskal-Wallis, Dunn	
				Postsmolt	3951.50	4361.31	2036.19	176.56	a	370.44–8352.18	126.15–8596.47	Kruskal-Wallis, Dunn	
				Adult	3812.90	4352.02	2502.36	347.02	a	428.48–12,130.70	174.0–12,743.0	Kruskal-Wallis, Dunn	
			Rainbow trout	Presmolt	2285.00	2713.33	1799.22	328.49	b	790.0–7500.0	790.0–7500.0	Kruskal-Wallis, Dunn	
				Smolt	3939.00	3890.86	2372.30	199.78	a	597.90–9265.55	351.80–12,833.83	Kruskal-Wallis, Dunn	
				Postsmolt	3306.85	3443.51	1808.86	137.92	a	578.61–7795.09	347.0–8608.0	Kruskal-Wallis, Dunn	
				Adult	4206.60	4258.64	2108.70	274.53	a	651.70–8941.30	189.0–8948.60	Kruskal-Wallis, Dunn	
Lactate dehydrogenase	LDH	U/L	Coho salmon	Smolt	775.00	781.61	284.63	25.46	a	194.81–1324.83	87.63–1424.18	Kruskal-Wallis, Dunn	
				Postsmolt	619.50	686.95	301.19	28.21	b	330.0–1572.38	292.0–1626.0	Kruskal-Wallis, Dunn	
				Adult	848.00	823.44	309.42	19.23	c	181.70–1405.37	113.53–1465.71	Kruskal-Wallis, Dunn	
			Atlantic salmon	Presmolt	980.00	1007.72	248.13	25.46	b	492.44–1488.65	407.86–1571.51	Kruskal-Wallis, Dunn	
				Smolt	1018.00	1122.48	446.28	39.76	b	431.75–2342.0	296.25–2496.50	Kruskal-Wallis, Dunn	
				Postsmolt	873.00	869.34	364.41	17.03	a	131.86–1564.84	60.42–1629.83	Kruskal-Wallis, Dunn	
				Adult	952.00	954.57	309.14	17.97	c	343.34–1559.91	258.42–1639.88	Kruskal-Wallis, Dunn	
			Rainbow trout	Presmolt	1130.00	1215.77	539.96	105.89	c	400.0–2140.0	400.0–2140.0	Kruskal-Wallis, Dunn	
				Smolt	733.00	835.04	392.65	27.03	a	65.46–1604.61	0.87–1669.21	Kruskal-Wallis, Dunn	
				Postsmolt	615.00	696.93	342.99	23.45	a	233.13–1615.0	174.0–1788.38	Kruskal-Wallis, Dunn	
				Adult	870.00	882.25	308.17	23.03	d	219.87–1440.16	155.0–1529.68	Kruskal-Wallis, Dunn	

**Table 6 biology-11-01066-t006:** Reference intervals (RIs) for plasma minerals and blood gasometry biomarkers and cortisol in presmolt and smolt (freshwater) and postsmolt and adult (seawater) of Atlantic salmon, coho salmon, and rainbow trout reared in Chile. The respective confidence intervals (CIs) for the respective RIs are included. Letters indicate significant differences between age ranges (*p* < 0.05).

Parameter	Abbreviation	Unit of Measure	Species	Age Stage	Median	Mean	S. D	S. E	Difference between Means	95% Reference Interval	Confidence Interval (95%)	Test	Data Processing
Sodium	Na	mmol/L	Coho salmon	Smolt	159.00	158.17	6.17	0.41	c	146.47–171.04	145.12–172.25	Kruskal-Wallis, Dunn	
				Postsmolt	171.05	170.99	4.16	0.39	a	162.67–179.22	160.93–181.0	Kruskal-Wallis, Dunn	
				Adult	166.00	164.66	9.26	0.54	b	147.67–184.12	145.97–185.55	Kruskal-Wallis, Dunn	
			Atlantic salmon	Presmolt	160.00	159.63	6.57	0.48	a	147.05–173.05	144.74–175.17	Kruskal-Wallis, Dunn	
				Smolt	170.00	167.33	8.83	0.48	b	150.02–186.37	148.58–187.42	Kruskal-Wallis, Dunn	
				Postsmolt	173.00	173.32	10.36	0.47	c	153.43–194.23	152.04–195.49	Kruskal-Wallis, Dunn	
				Adult	180.00	176.93	10.62	0.57	d	158.55–202.07	156.05–203.90	Kruskal-Wallis, Dunn	
			Rainbow trout	Presmolt	179.90	179.40	3.30	0.62	a	173.09–187.06	170.64–188.46	Kruskal-Wallis, Dunn	
				Smolt	157.00	156.01	10.45	0.63	c	135.89–177.28	133.89–179.17	Kruskal-Wallis, Dunn	
				Postsmolt	156.00	156.12	13.10	0.85	cb	130.37–182.11	127.89–184.51	Kruskal-Wallis, Dunn	
				Adult	168.00	164.61	12.61	0.87	b	142.03–193.58	138.60–197.61	Kruskal-Wallis, Dunn	
Potassium	K	mmol/L	Coho salmon	Smolt	3.40	4.08	2.69	0.18	b	1.17–10.49	1.0–12.30	Kruskal-Wallis, Dunn	
				Postsmolt	0.99	1.17	0.72	0.07	c	0.53–3.85	0.45–5.21	Kruskal-Wallis, Dunn	
				Adult	4.40	5.02	2.86	0.17	a	1.20–13.28	1.10–14.26	Kruskal-Wallis, Dunn	
			Atlantic salmon	Presmolt	3.70	3.90	1.95	0.14	b	0.83–8.76	0.45–9.29	Kruskal-Wallis, Dunn	
				Smolt	3.60	3.69	1.48	0.08	b	0.59–6.45	0.29–6.75	Kruskal-Wallis, Dunn	
				Postsmolt	3.00	3.37	1.89	0.09	a	0.87–8.32	0.80–9.60	Kruskal-Wallis, Dunn	
				Adult	4.30	4.50	2.25	0.12	c	1.07–9.73	0.90–10.60	Kruskal-Wallis, Dunn	
			Rainbow trout	Presmolt	2.80	2.78	1.23	0.23	bc	0.76–6.02	0.76–6.02	Kruskal-Wallis, Dunn	
				Smolt	2.30	2.81	1.95	0.12	c	0.70–8.57	0.70–8.88	Kruskal-Wallis, Dunn	
				Postsmolt	3.90	4.29	2.06	0.13	a	1.20–9.50	1.10–10.28	Kruskal-Wallis, Dunn	
				Adult	3.70	3.62	1.58	0.11	b	0.80–8.17	0.70–8.80	Kruskal-Wallis, Dunn	
Chloride	Cl	mmol/L	Coho salmon	Smolt	130.85	130.38	11.12	0.73	c	108.90–152.87	106.78–154.61	Kruskal-Wallis, Dunn	
				Postsmolt	141.25	141.92	4.38	0.41	a	132.66–150.27	130.68–152.12	Kruskal-Wallis, Dunn	
				Adult	139.25	134.26	13.13	0.75	b	109.23–164.61	106.51–166.66	Kruskal-Wallis, Dunn	
			Atlantic salmon	Presmolt	119.00	120.09	6.91	0.50	a	105.85–133.50	104.30–135.09	Kruskal-Wallis, Dunn	
				Smolt	125.70	127.38	12.08	0.67	a	101.58–149.61	99.85–152.02	Kruskal-Wallis, Dunn	
				Postsmolt	134.05	138.07	12.43	0.57	c	111.44–162.83	110.17–164.46	Kruskal-Wallis, Dunn	
				Adult	143.10	143.17	12.57	0.67	d	118.62–168.17	116.8–169.81	Kruskal-Wallis, Dunn	
			Rainbow trout	Presmolt	141.80	142.16	2.04	0.39	a	138.16–146.16	137.12–147.21	ANOVA, Tukey	BOX COX
				Smolt	120.60	122.58	12.47	0.76	c	97.30–147.14	94.96–149.08	ANOVA, Tukey	BOX COX
				Postsmolt	119.00	120.26	9.23	0.60	c	101.03–137.83	99.28–139.80	ANOVA, Tukey	BOX COX
				Adult	131.25	129.03	15.07	1.04	b	99.68–159.91	96.72–162.55	ANOVA, Tukey	BOX COX
Calcium	Ca	mmol/L	Coho salmon	Smolt	8.75	8.91	1.58	0.21	a	5.66–12.08	5.14–12.64	ANOVA, Tukey	Normal
				Postsmolt	12.25	12.12	1.56	0.15	c	9.31–15.52	8.81–16.02	ANOVA, Tukey	Normal
				Adult	13.56	13.25	2.96	0.51	b	8.64–18.98	7.34–20.22	ANOVA, Tukey	Normal
			Atlantic salmon	Presmolt	14.45	14.78	1.66	0.26	a	11.34–17.53	10.57–18.45	ANOVA, Tukey	POWER BOX COX
				Smolt	11.60	11.68	1.44	0.19	b	8.61–14.41	8.06–15.04	ANOVA, Tukey	POWER BOX COX
				Postsmolt	11.96	12.13	2.14	0.29	b	8.22–16.33	7.60–17.08	ANOVA, Tukey	POWER BOX COX
				Adult	12.09	12.04	1.40	0.26	b	9.09–14.87	8.16–15.69	ANOVA, Tukey	POWER BOX COX
			Rainbow trout	Presmolt	16.99	15.78	2.83	0.57	a	9.73–22.72	7.89–24.36	ANOVA, Tukey	BOX COX
				Smolt	13.32	13.33	1.96	0.21	b	9.33–17.15	8.83–17.80	ANOVA, Tukey	BOX COX
Magnesium	Mg	mmol/L	Coho salmon	Smolt	4.60	4.56	1.12	0.15	a	2.26–6.81	1.89–7.22	ANOVA, Tukey	BOX COX
				Postsmolt	1.31	1.33	0.32	0.03	b	0.68–1.97	0.60–2.05	ANOVA, Tukey	BOX COX
				Adult	1.06	1.15	0.26	0.04	c	0.51–1.63	0.39–1.80	ANOVA, Tukey	BOX COX
			Atlantic salmon	Presmolt	3.46	3.56	0.61	0.11	b	2.17–4.71	1.73–5.14	ANOVA, Tukey	BOX COX
				Smolt	6.25	5.96	1.12	0.15	a	3.70–8.39	3.27–8.77	ANOVA, Tukey	BOX COX
				Postsmolt	5.36	5.75	1.45	0.22	a	2.58–8.75	1.59–9.43	ANOVA, Tukey	BOX COX
				Adult	2.94	3.24	0.81	0.15	b	1.40–5.02	0.99–5.44	ANOVA, Tukey	BOX COX
			Rainbow trout	Presmolt	3.70	3.81	0.85	0.16	a	2.0–5.62	1.66–5.98	ANOVA, Tukey	BOX COX
				Smolt	3.31	3.33	0.44	0.05	b	2.43–4.21	2.26–4.36	ANOVA, Tukey	BOX COX
				Postsmolt	1.57	1.57	0.23	0.04	c	1.10–2.05	0.97–2.17	ANOVA, Tukey	BOX COX
Iron	Fe	μmol/L	Coho salmon	Smolt	72.50	108.86	73.56	10.62	a	26.95–244.50	26.50–244.40	ANOVA, Tukey	POWER BOX COX
				Postsmolt	58.30	64.27	25.30	2.41	c	25.12–131.99	17.30–170.0	ANOVA, Tukey	POWER BOX COX
				Adult	103.00	106.99	30.76	5.20	b	45.58–169.08	31.28–186.60	ANOVA, Tukey	POWER BOX COX
			Atlantic salmon	Presmolt	125.15	122.42	26.37	4.81	c	87.09–159.23	76.95–171.46	ANOVA, Tukey	POWER BOX COX
				Smolt	114.50	119.62	31.60	4.08	b	51.04–180.10	37.43–193.84	ANOVA, Tukey	POWER BOX COX
				Postsmolt	75.60	70.81	19.87	2.61	a	34.31–111.09	27.83–116.32	ANOVA, Tukey	POWER BOX COX
			Rainbow trout	Presmolt	6.93	8.19	4.26	0.78	b	1.97–17.09	1.97–17.09	ANOVA, Tukey	Normal
				Smolt	13.45	13.43	4.06	0.44	a	5.37–21.59	4.02–22.97	ANOVA, Tukey	Normal
				Postsmolt	10.67	11.34	3.07	0.57	a	4.58–17.72	2.85–19.49	ANOVA, Tukey	Normal
Phosphorus	P	mmol/L	Coho salmon	Smolt	29.00	29.38	3.65	0.48	a	21.90–36.76	20.77–38.09	ANOVA, Tukey	sqrt(x)
				Postsmolt	16.39	16.71	2.35	0.22	c	11.90–21.34	11.32–21.96	ANOVA, Tukey	sqrt(x)
				Adult	20.86	20.11	3.42	0.59	b	13.05–27.53	11.29–28.99	ANOVA, Tukey	sqrt(x)
			Atlantic salmon	Presmolt	21.58	21.37	2.85	0.37	c	15.86–27.40	14.65–28.32	ANOVA, Tukey	POWER BOX COX
				Smolt	28.25	26.67	7.45	1.01	b	11.54–42.48	8.83–45.66	ANOVA, Tukey	POWER BOX COX
				Postsmolt	30.39	30.79	3.14	0.41	a	24.01–36.78	22.92–38.16	ANOVA, Tukey	POWER BOX COX
				Adult	11.72	11.69	1.81	0.33	d	7.87–15.36	6.65–16.34	ANOVA, Tukey	POWER BOX COX
			Rainbow trout	Presmolt	23.15	23.86	3.68	0.67	a	15.85–31.51	13.91–33.41	ANOVA, Tukey	POWER BOX COX
				Smolt	21.28	21.85	4.35	0.46	b	12.69–30.22	11.22–31.78	ANOVA, Tukey	POWER BOX COX
				Postsmolt	24.68	25.07	3.17	0.58	a	18.29–31.59	16.75–33.38	ANOVA, Tukey	POWER BOX COX
Bicarbonate ion concentration	HCO_3_^−^	mmol/L	Coho salmon	Postsmolt	7.43	7.43	1.12	0.14	NA	5.15–9.65	4.78–10.05	NA	
			Atlantic salmon	Presmolt	10.71	10.86	1.37	0.18	a	7.86–13.44	7.21–14.12	ANOVA, Tukey	log(x + 1)
				Postsmolt	8.41	8.53	0.90	0.13	b	6.60–10.28	6.31–10.69	ANOVA, Tukey	log(x + 1)
			Rainbow trout	Presmolt	10.92	11.24	1.39	0.26	a	8.17–14.15	7.57–14.89	ANOVA, Tukey	Normal
				Smolt	7.72	7.61	1.35	0.26	b	4.86–10.55	4.01–11.35	ANOVA, Tukey	Normal
Partial pressure of carbon dioxide	pCO_2_	mmHg	Coho salmon	Postsmolt	11.90	12.50	2.31	0.28	NA	7.32–16.92	6.65–17.86	NA	
			Atlantic salmon	Presmolt	22.55	22.76	2.67	0.36	a	17.33–28.19	16.41–29.08	Kruskal-Wallis, Dunn	
				Postsmolt	9.20	7.85	3.58	0.53	b	1.67–12.88	1.62–13.0	Kruskal-Wallis, Dunn	
			Rainbow trout	Presmolt	20.80	20.74	1.72	0.33	a	17.19–24.43	16.23–25.23	ANOVA, Tukey	Normal
				Smolt	16.50	16.94	1.57	0.31	b	13.41–20.29	12.53–21.18	ANOVA, Tukey	Normal
Hydrogen potential	pH		Coho salmon	Postsmolt	7.39	7.39	0.06	0.01	NA	7.28–7.54	7.28–7.55	NA	
			Atlantic salmon	Presmolt	7.29	7.29	0.08	0.01	a	7.12–7.45	7.09–7.48	ANOVA, Tukey	Normal
				Postsmolt	7.32	7.32	0.12	0.02	a	7.08–7.55	7.04–7.60	ANOVA, Tukey	Normal
			Rainbow trout	Presmolt	7.34	7.34	0.07	0.01	a	7.20–7.48	7.17–7.52	ANOVA, Tukey	Normal
				Smolt	7.28	7.26	0.10	0.02	b	7.06–7.48	7.0–7.53	ANOVA, Tukey	Normal
Cortisol	COR	ng/mL	Coho salmon	Smolt	34.69	27.23	26.6	1.80	b	8.22–92.97	4.71–117.17	ANOVA, Tukey	POWER BOX COX
				Postsmolt	77.65	77.64	37.3	5.63	a	13.35–129.91	3.32–134.37	ANOVA, Tukey	POWER BOX COX
				Adult	30.90	27.68	15.1	0.90	b	10.52–64.0	10.07–69.17	ANOVA, Tukey	POWER BOX COX
			Atlantic salmon	Presmolt	23.82	19.20	15.8	1.76	c	9.60–99.39	2.17–71.74	Kruskal-Wallis, Dunn	Not normal
				Smolt	39.28	37.85	20.5	1.17	b	11.44–114.71	7.90–93.26	Kruskal-Wallis, Dunn	Not normal
				Postsmolt	55.60	50.40	31.3	1.54	a	6.28–105.12	11.28–140.83	Kruskal-Wallis, Dunn	Not normal
				Adult	59.10	56.60	28.2	1.52	a	4.23–59.49	10.4–118.80	Kruskal-Wallis, Dunn	Not normal
			Rainbow trout	Presmolt	37.81	34.94	15.4	2.82	a	6.90–75.18	7.28–72.75	Kruskal-Wallis, Dunn	Not normal
				Smolt	39.17	35.48	22.0	1.51	a	7.28–72.75	7.47–100.47	Kruskal-Wallis, Dunn	Not normal
				Postsmolt	43.26	40.76	19.1	2.25	a	4.19–134.14	8.53–102.17	Kruskal-Wallis, Dunn	Not normal
				Adult	30.33	22.57	21.42	1.83	b	10.87–88.47	6.52–94.49	Kruskal-Wallis, Dunn	Not normal

## Data Availability

The data presented in this study are available on request from the corresponding author.

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
