# Peer review of "Reference Intervals for Blood Biomarkers in Farmed Atlantic Salmon, Coho Salmon and Rainbow Trout in Chile: Promoting a Preventive Approach in Aquamedicine"

_biology, 2022, doi:10.3390/biology11071066_

Round 1

Reviewer 1 Report

The authors present novel and interesting information about reference intervals for blood biomarkers in three fish species farmed in Chile: Atlantic salmon, coho salmon and rainbow trout. Overall, the manuscript is well written and presented. I think this paper would be of great interest to the readers of Biology. Therefore, I would recommend this manuscript for publication after minor changes and comments are addressed, mainly in the material and methods section.

Specific comments:

Abstract: No comments

Introduction:

Line 81 and 82: Reference as author (year)

Line 106 and 107: Please use consistency in units for values presented, here you used mg L-1 and mg/L. Choose the format that is required by the journal.

Materials and Methods:

Section 2.1.2: This section refers to the anaesthetic procedure exerted on the fish, however, there is no indication of how the fish were kept or collected in the holding tanks before the application of the anaesthetic. Were the fish captured using nets? Any other method, manipulation? Was the same capture method used in all the 78 fish farms? All these details need to be clearly described as reference ranges of biomarkers could be affected by different capture methods.

Line 171: What portable equipment? This is the first mention of this equipment so please explain what this equipment is or does.   

Line 174: …non-vacuum sealed blood collection tube? containing…

Line 176: Type of tube? how big? You mention a mark in the tube but do not specify the volume.

Line 181: Were syringes also placed in polystyrene boxes together with the tubes containing the blood samples?

Line 184: labelled

Line 187: I would re-name this section to “Analytical tests”

Line 190: Is this the portable equipment mentioned in Line 171?

Line 194: Please describe the protocol for the RBC and WBC counts using the Natt-Herrick staining solution. What was the mixing ratio of blood and staining solution (v:v)?

Line 212: Please briefly describe the method described by Wintrobe (1934) to calculate MCV and MCHC.

Section 2.2.2: Please mention in this section the specific plasma substrates, enzymes, electrolytes and minerals analysed, including their acronym. As it currently stands, the first time the reader sees which particular parameters were tested is in the Results section (section 3.3) and it is only the acronyms.

Line 217-219: Please describe which kits were used for all the parameters tested and a brief description of the method for each test.

Line 220: I would re-name this section to “Statistical Analyses”

Line 237: Reference as author (year)

Results:

Line 341, 343, 388: Please add the name of the tested markers to the methods section

Line 442: please add the name of all tested plasma electrolytes and minerals to the methods section

Discussion:

Nice short and focused discussion. The only comment I have relates to Line 530-531 where you mention that blood biochemistry is affected by intrinsic and extrinsic factors such as population density and catching methods. Yet, in your methods, there is no indication of how the sampled fish were held in the tanks or how they were captured for sampling.  

Figures & Tables:

All figures: the legends are very small and it makes them almost impossible to read. Also, is there a reason why panel C has a border in all figures and not A and B? For consistency purposes, I would recommend either using a border in all panels of each figure or none at all.

Tables 1-6: in some columns in all these tables, a decimal comma has been used as a decimal separator. Please correct it and use a decimal point instead as I am sure this is the journal’s guideline.

General comments:

Please make sure you use subscript consistently when refereeing to pCO2 or HCO3 throughout the text. These are used correctly in figure legends, but not in the main text. 

Author Response

Specific comments:

Abstract: No comments

Introduction: 

Line 81 and 82: Reference as author (year)

R: The format of the references has been modified in accordance with MDPI guidelines.

Line 106 and 107: Please use consistency in units for values presented, here you used mg L-1and mg/L. Choose the format that is required by the journal.

R: The format of the references has been modified in accordance with MDPI guidelines.

Materials and Methods:

Section 2.1.2: This section refers to the anaesthetic procedure exerted on the fish, however, there is no indication of how the fish were kept or collected in the holding tanks before the application of the anaesthetic. Were the fish captured using nets? Any other method, manipulation? Was the same capture method used in all the 78 fish farms? All these details need to be clearly described as reference ranges of biomarkers could be affected by different capture methods.

R: Good point. thank you very much. We have included the following paragraph L136-L142: “All freshwater-reared fish (presmolt and smolts) were captured using the same protocol, regardless of salmon producer and hatchery. Briefly, no more than five fish were collected at the same time directly from each selected tank using a small fishing net with handle (locally referred to as a "quecha") and then quickly deposited into the bucket with anesthesia. In addition, all sea-reared fish (postsmolt and adults) were also captured using the same protocol between seawater farms, but fish were captured by the crowd and net method.”

Line 171: What portable equipment? This is the first mention of this equipment so please explain what this equipment is or does.   

R: This section is only to describe the sample collection (pre-analytical), so the analytical protocol and the detail of the portable blood gas equipment used is described in the corresponding point (2.2.1) in L182.

Line 174: …non-vacuum sealed blood collection tube? containing…

R: The word "tube" has been included and the line now reads as follows (L167): “Whole blood samples for hematological and blood biochemistry tests were collected in a volume that varied from 1 to 3 mL from the caudal vein of each fish using non-vacuum sealed blood collection tube containing lithium heparin”

Line 176: Type of tube? how big? You mention a mark in the tube but do not specify the volume.

R: This was detailed shortly before, at L166, but the volume varied between 1 to 3 ml depending on the size of the fish.

Line 181: Were syringes also placed in polystyrene boxes together with the tubes containing the blood samples?

R: No. Blood samples for blood gas analysis were taken in the field and analyzed in the field with the portable equipment. They were not taken to the laboratory.

Line 184: labelled

The word "labelled" has been included and the line now reads as follows (L174)

Line 187: I would re-name this section to “Analytical tests”

R: The methodology was presented according to the critical stages for the execution of blood analysis: pre-analytical stage; analytical stage and post-analytical stage. Obviously, the tests performed fall in the analytical stage and the statistical analyses in the post-analytical stage. In this way, the critical points of the methodology are more clearly exposed.

Line 190: Is this the portable equipment mentioned in Line 171?

R: That is correct.

Line 194: Please describe the protocol for the RBC and WBC counts using the Natt-Herrick staining solution. What was the mixing ratio of blood and staining solution (v:v)?

R: We believe that it is not necessary to fill the methodology with the details of protocols that are already very well standardized, validated and profusely described in the literature, in order to focus efforts on what is important.

Line 212: Please briefly describe the method described by Wintrobe (1934) to calculate MCV and MCHC.

R: Idem.

Section 2.2.2: Please mention in this section the specific plasma substrates, enzymes, electrolytes and minerals analysed, including their acronym. As it currently stands, the first time the reader sees which parameters were tested is in the Results section (section 3.3) and it is only the acronyms.

R: We have included each biomarker that was analyzed directly in the text and the lines now read as follows:

L182-L183: “The blood gas biomarkers analyzed were bicarbonate ion concentration (HCO3), partial pressure of carbon dioxide (pCO2), and hydrogen potential (pH).”

L194-L197: “The blood count biomarkers analyzed were hematocrit (Htc), red blood cell count (RBC), hemoglobin (Hgb), mean corpuscular volume (MCV), mean corpuscular hemoglobin concentration (MCHC), white blood cell count (WBC), lymphocytes (LYM), neutrophils (NEU), monocytes (MON) and thrombocyte count (TCC).”

L212-L222: “[Total protein (TPO), Albumins (ALB), Globulins (GLO), Total bilirubin (TBI), Direct bilirubin (DBI), Creatinine (CRE), Glucose (GLU), Lactate (LAC), Urea (URE), Uric acid (UAC), Amonio (NH3), Total Cholesterol (TCH), Triglycerides (TRG), High-density lipoprotein cholesterol (HDL), and Low-density lipoprotein cholesterol (LDL)], enzymes [Alkaline Phosphatase (ALP), Alanine transaminase (ALT), Aspartate aminotransferase (AST), Total amylase (TAM), Lipase (LIP), Creatine Kinase total (CKT), Cardiac Creatine Kinase isoenzyme (CK-MB), and Lactate dehydrogenase (LDH)], electrolytes and minerals [Sodium (Na), Potassium (K), Chloride (Cl), Calcium (Ca), Magnesium (Mg), Iron (Fe), Phosphorus (P)] using a cobas c311 autoanalyzer (Roche Diagnostics, Risch-Rotkreuz, Switzerland), while plasma cortisol (COR)…”

Line 217-219: Please describe which kits were used for all the parameters tested and a brief description of the method for each test.

R: We believe that it is not necessary to fill the methodology with the details of protocols that are already very well standardized, validated and profusely described in the literature, in order to focus efforts on what is important.

Line 220: I would re-name this section to “Statistical Analyses”

R: The methodology was presented according to the critical stages for the execution of blood analysis: pre-analytical stage; analytical stage and post-analytical stage. Obviously, the tests performed fall in the analytical stage and the statistical analyses in the post-analytical stage. In this way, the critical points of the methodology are more clearly exposed.

Line 237: Reference as author (year)

R: The format of the references has been modified in accordance with MDPI guidelines.

Results:

Line 341, 343, 388: Please add the name of the tested markers to the methods section

R: OK. It’s done.

Line 442: please add the name of all tested plasma electrolytes and minerals to the methods section.

R: OK. It’s done.

Discussion:

Nice short and focused discussion. The only comment I have relates to Line 530-531 where you mention that blood biochemistry is affected by intrinsic and extrinsic factors such as population density and catching methods. Yet, in your methods, there is no indication of how the sampled fish were held in the tanks or how they were captured for sampling.

R: OK. It’s done. 

Figures & Tables:

All figures: the legends are very small, and it makes them almost impossible to read. Also, is there a reason why panel C has a border in all figures and not A and B? For consistency purposes, I would recommend either using a border in all panels of each figure or none at all.

Tables 1-6: in some columns in all these tables, a decimal comma has been used as a decimal separator. Please correct it and use a decimal point instead as I am sure this is the journal’s guideline.

R: OK. It’s done.

General comments:

Please make sure you use subscript consistently when refereeing to pCO2 or HCO3 throughout the text. These are used correctly in figure legends, but not in the main text.

R: OK. It’s done.

Reviewer 2 Report

In this article entitled “Reference intervals for blood biomarkers in farmed Atlantic salmon, coho salmon and rainbow trout in Chile: promoting a preventive approach in aquamedicine” we find detailed Reference Intervals (RIs) of several essential biomarkers to know the health status of different species of salmonids collected throughout different age ranges. The selection of samples for this work seems to be very appropriate since it collects a large number of individuals distributed by 78 different fish farms throughout Chile, covering great distances with their consequent variability to obtain more reliable RIs.

One of the weaknesses that I observe in the article is the introduction, since certain phrases like the one that goes from line 65 to 70 seem to be meaningless, the reason for studying these cells should be better clarified.

Another thing that I cannot handled is the order and form of the citations throughout the text, since they do not consist of any type of order, I strongly urge you to review the journal's standards for the format of bibliographic citations.

In the section 2.1.3  should be clarified if the blood for the gasometry and the blood for the hematological and blood biochemistry tests were obtained from the same fish. One thing that is not clear is whether absolutely all these variables are measured in each fish (the samples being sufficient for all the tests) or whether not enough sample is obtained from some fish (for example, the youngest) to carry out the analysis of all biomarkers.

In the line 195 is it really necessary to explain that the samples are thrown in a container and that the printed data is obtained? It stands to reason that we have to properly discard the syringes and not used samples and it stands to reason that the data can be printed.

Another thing that is not clear is whether these data would be reproducible in another laboratory, since after arguing throughout the text that your data will be essential to set the RIs for these 3 fish species and thus know the health status of them, in line 576 is said “so each diagnostic laboratory should generate the RIs based on their own methods and equipment”. To my understanding, this phrase ruins the fact that, thanks to your article, reference intervals are described and will be useful for any other laboratory.

Author Response

One of the weaknesses that I observe in the article is the introduction, since certain phrases like the one that goes from line 65 to 70 seem to be meaningless, the reason for studying these cells should be better clarified.

R: Thank you very much for your comment, but we missed your point. The main objective of our study is clear and correctly supported. If you can be a little more specific, maybe we can understand your point correctly.

Another thing that I cannot handled is the order and form of the citations throughout the text, since they do not consist of any type of order, I strongly urge you to review the journal's standards for the format of bibliographic citations.

R: The format of the references has been modified in accordance with MDPI guidelines.

In the section 2.1.3  should be clarified if the blood for the gasometry and the blood for the hematological and blood biochemistry tests were obtained from the same fish. One thing that is not clear is whether absolutely all these variables are measured in each fish (the samples being sufficient for all the tests) or whether not enough sample is obtained from some fish (for example, the youngest) to carry out the analysis of all biomarkers.

R: OK. It’s done.

In the line 195 is it really necessary to explain that the samples are thrown in a container and that the printed data is obtained? It stands to reason that we have to properly discard the syringes and not used samples and it stands to reason that the data can be printed.

R: It is necessary because standardized pre-analytical protocols are proposed for users in field conditions, technical-operational persons or professionals who are not necessarily well-trained persons for the execution of this protocol. Therefore, it is a matter of personnel training as well.

Another thing that is not clear is whether these data would be reproducible in another laboratory, since after arguing throughout the text that your data will be essential to set the RIs for these 3 fish species and thus know the health status of them, in line 576 is said “so each diagnostic laboratory should generate the RIs based on their own methods and equipment”. To my understanding, this phrase ruins the fact that, thanks to your article, reference intervals are described and will be useful for any other laboratory.

R: What we are saying is that the RIs we report here were estimated according to the conditions exposed in our study, which are useful for all that is indicated, but that (conditionally and in an ideal scenario), each laboratory should estimate its RIs due to all that is discussed in the work. As we also indicated, doing the latter determines a high economic cost, so historically the RIs defined in the literature have been used. As our study is the most complete so far reported in salmonids, we estimate that it will be a material that other laboratories and researchers will have as a reference.
